# WHY FINE-TUNING STRUGGLES WITH FORGETTING IN MACHINE UNLEARNING? THEORETICAL INSIGHTS AND A REMEDIAL APPROACH

## ABSTRACT

Machine Unlearning has emerged as a significant area of research, focusing on 'removing' specific subsets of data from a trained model. Fine-tuning (FT) methods have become one of the fundamental approaches for approximating unlearning, as they effectively retain model performance. However, it is consistently observed that naive FT methods struggle to forget the targeted data. In this paper, we present the first theoretical analysis of FT methods for machine unlearning within a linear regression framework, providing a deeper exploration of this phenomenon. We investigate two scenarios with distinct features and overlapping features. Our findings reveal that FT models can achieve zero remaining loss yet fail to forget the forgetting data, unlike golden models (trained from scratch without the forgetting data). This analysis reveals that naive FT methods struggle with forgetting because the pretrained model retains information about the forgetting data, and the fine-tuning process has no impact on this retained information. To address this issue, we first propose a theoretical approach to mitigate the retention of forgetting data in the pretrained model. Our analysis shows that removing the forgetting data's influence allows FT models to match the performance of the golden model. Building on this insight, we revisit the discriminative regularization used in existing studies and redesign it to effectively reduce the unlearning loss gap between the fine-tuned model and the golden model. Our experiments on both synthetic and real-world datasets validate these theoretical insights and demonstrate the effectiveness of the advanced regularization method.

## 1 INTRODUCTION

Machine Unlearning has emerged as a prominent area that focuses on protecting individual privacy during the model training process, particularly adhering to legislation such as 'the right to be forgotten' (Rosen, 2011) under the General Data Protection Regulation (GDPR) (Hoofnagle et al., 2019). That is, it removes certain training samples from the trained model upon their users' data deletion request. A natural approach to machine unlearning is to retrain the model from scratch, excluding the data that needs to be forgotten; this is known as exact unlearning. However, this method is highly computationally inefficient. To address this challenge, previous research has proposed a more relaxed definition of machine unlearning, where the unlearned model only needs to be approximately similar to one retrained from scratch. This led to the development of *approximate unlearning* methods, such as Fine-Tuning (Warnecke et al., 2021; Golatkar et al., 2020a), Gradient Ascent (Graves et al., 2021; Thudi et al., 2022), Fisher Forgetting (Becker & Liebig, 2022; Golatkar et al., 2020a), and Influence Unlearning(Izzo et al., 2021).

Fine-tuning, as one of the most widely used approaches in approximate unlearning, has demonstrated its empirical effectiveness. However, it can be observed in many studies (Kurmanji et al., 2024; Warnecke et al., 2021; Golatkar et al., 2020a; Liu et al., 2024; Sharma et al., 2024) and our investigations in Table 1 that while fine-tuning may maintain the utility of the model on remaining data, it struggles to forget the targeted data. This raises a natural question:

*Why does fine-tuning fail to unlearn the forgetting data in machine unlearning?*

Table 1: Cifar-10 Class-wise Forgetting Performance Comparing Retrain and Naive FT (Fine-Tuning) Method. The table compares Retrain and FT on CIFAR-10 across multiple evaluation metrics: Unlearning Accuracy (UA), Retaining Accuracy (RA), MIA-Efficacy, Test Accuracy (TA), and Run-Time. Values in brackets indicate the gap between FT and the golden model (i.e., Retrain). Further explanations are provided in Section 6.

| Methods | UA | RA | Cifar-10 Class-wise Forgetting MIA-Efficacy | TA | Run Time |
|---------|-----|-----|------|-----|-----|
| Retrain | $100.00_{\pm 0.00}$ | $100.00_{\pm 0.00}$ | $100.00_{\pm 0.00}$ | $94.87_{\pm 0.14}$ | 77.00 |
| FT | $20.89_{\pm 4.12}(79.11)$ | $99.76_{\pm 0.12}(0.24)$ | $74.32_{\pm 11.90}(25.68)$ | $93.73_{\pm 0.22}(1.14)$ | 4.48 |

To answer this question, we revisit the machine unlearning problem with a simple yet fundamental over-parameterized linear regression model and explore the behavior of fine-tuning through a theoretical perspective. Our main contributions can be summarized as follows.

- **Theoretical Analysis: Distinct and Overlapping Features.** We provide the first theoretical analysis of FT methods in the context of machine unlearning within a linear regression framework. Specifically, **1**) Based on the assumption of distinct features (Assumption 3.1), our theoretical observations, which align with empirical studies, show that the remaining loss for the fine-tuning model is zero, matching that of the golden model. Moreover, the loss of the fine-tuning model on the forgetting dataset consistently remains zero, diverging from the performance of the golden model. **2**) we extend our analysis to a more complex case when the dataset retained for model retraining shares overlapping features with the forgetting dataset. This challenges assumptions of distinct feature sets across datasets, yet the previous conclusions remain valid in this case. More discussion refers to Section 3.

- **Theory for Enhanced Unlearning.** Our analysis shows that naive fine-tuning (FT) methods fail to unlearn the forgetting data because the pretrained model retains information about this data, and the fine-tuning process does not effectively alter that retention. To address this issue, we propose a theoretical approach that removes the influence of the forgetting data, mitigating its retention in the pretrained model. This enables FT models to significantly improve unlearning accuracy while preserving the accuracy on the remaining data. Furthermore, our findings provide key insights for designing machine unlearning algorithms: retaining overlapping features between the remaining and forgetting datasets has minimal impact on unlearning accuracy, while discarding these features results in a decrease in the accuracy on the remaining data.

- **Revisiting Discriminative Regularization.** We revisit the discriminative regularization used in existing studies and redesign it to shift the regularization focus, prioritizing retaining accuracy over unlearning accuracy. This shift ensures that overlapping features between the forgetting and remaining datasets are preserved, maintaining overall model utility. Moreover, we incorporate KL-divergence loss alongside cross-entropy to better capture the distributional discrepancies for effective unlearning.

- **Experimental Validation on Synthetic and Real-World Data.** We validate our theoretical findings on both synthetic and real-world datasets. Firstly, our experiments demonstrate that all regularization-based methods significantly improve UA compared to the baseline FT. Furthermore, we observe that in the fine-tuning process, focusing on preserving accuracy on remaining data along with regularization on forgetting data to enhance unlearning will achieves both good RA and UA. However, placing more emphasis on using the forgetting data to improve UA can significantly degrade RA, consistent with our analysis.

## 1.1 RELATED WORK

**Machine Unlearning Methods.** Cao & Yang (2015) first defined "Unlearning" as the removal of a sample that produces the same output on the dataset as if the sample had never been trained. The natural way to solve the problem is to retrain a model from scratch in response to each data deletion request. However, retraining is not feasible due to the limited time and constrained resources. Ginart et al. (2019) provided a relaxed definition inspired by Differential Privacy (Dwork et al., 2014), which only requires the unlearned model to produce results similar to those of retrain-from-scratch models. This led to the development of "approximate unlearning" methods, offering

more efficient computational designs for machine unlearning. Guo et al. (2019); Izzo et al. (2021); Neel et al. (2021); Ullah et al. (2021); Sekhari et al. (2021) provide theoretical error guarantees by focusing on the empirical risk minimization problem under this probabilistic notion of unlearning. Golatkar et al. (2020a) proposed an information-based procedure to remove knowledge from the trained weights, without access to the original training data. Further, Golatkar et al. (2020b) approximated the weights inspired by NTK theory, addressing situations where the Hessian is not informative about where the model will converge into a null space. Mehta et al. (2022) avoid the computation of Hessian by introducing a method only computing conditional independence, which identifies the Markov Blanket of parameters requiring updates. Thudi et al. (2022) proposed a regularizer to reduce the 'verification error,' which represents the distance between the unlearned model and a retrained-from-scratch model. Kurmanji et al. (2024) bears a novel teacher-student formulation to achieve better performance towards unbounded unlearning problems. Liu et al. (2024) considers model sparsity by pruning weights before the unlearning process, thereby introducing a new unlearning paradigm. Shen et al. (2024) incorporates the variational inference and contrastive learning approaches to address the lack of supervision information (label-agnostic).

**Machine Unlearning Theory.** For approximate unlearning, Neel et al. (2021); Thudi et al. (2022) explored algorithms for empirical risk minimization objectives, while Sekhari et al. (2021) studied population risk minimization problems, providing theoretical guarantees on both the effectiveness of unlearning and the privacy of the data subjects. Guo et al. (2019); Zhang et al. (2022) provided the certified radius with respect to data changes before and after removals, as well as the certified budget for data removals. For exact unlearning, Ullah et al. (2021) introduced the notion of algorithmic stability, called Total Variation (TV) stability, which is suited for achieving exact unlearning. This concept was further extended to the federated setting by Che et al. (2023); Tao et al. (2024). However, existing theoretical work has primarily focused on utility guarantees, with limited analysis explaining the successes and failures of fine-tuning methods.

**Notations**: In this paper, we adhere to a consistent notation style for clarity. We use boldface lower letters such as $\mathbf{x}, \mathbf{w}$ for vectors, and boldface capital letters (e.g. $\mathbf{A}, \mathbf{H}$) for matrices. Let $\|\mathbf{A}\|_2$ denote the spectral norm of $\mathbf{A}$ and $\|\mathbf{v}\|_2$ denote the Euclidean norm of $\mathbf{v}$. For two vectors $\mathbf{u}$ and $\mathbf{v}$, their inner product is denoted by $\langle \mathbf{u}, \mathbf{v} \rangle$ or $\mathbf{u}^\top \mathbf{v}$. For two matrices $\mathbf{A}$ and $\mathbf{B}$ of appropriate dimension, their inner product is defined as $\langle \mathbf{A}, \mathbf{B} \rangle := \mathrm{tr}(\mathbf{A}^\top \mathbf{B})$. For a positive semi-definite (PSD) matrix $\mathbf{A}$ and a vector $\mathbf{v}$ of appropriate dimension, we write $\|\mathbf{v}\|_{\mathbf{A}}^2 := \mathbf{v}^\top \mathbf{A} \mathbf{v}$. Denote by $\mathbf{P}_m$ the projection onto the space of a matrix $\mathbf{X}_m$, i.e., $\mathbf{P}_m = \mathbf{X}_m (\mathbf{X}_m^\top \mathbf{X}_m)^{-1} \mathbf{X}_m^\top$.

## 2 MACHINE UNLEARNING IN LINEAR MODELS

Let $D = \{(\mathbf{x}_i, y_i)\}_{i=1}^n$ be a training dataset consisting of $n$ data points, where $\mathbf{x}_i$ represents the feature vector, and $y_i$ is the response variable for each data point in the dataset $D$. Assume that each pair $(\mathbf{x}_i, y_i)$ is a realization of the linear regression model: $y = \mathbf{x}^\top \mathbf{w}_*$, with $\mathbf{w}_* \in \mathbb{R}^d$ being the optimal model parameter in the overparameterized regime ($n \ll d$). Machine Unlearning aims to remove (or scrub) the influence of specific training data from a trained machine learning (ML) model. Let $D_f = \{(\mathbf{x}_i, y_i)\}_{i=1}^{n_f} \subseteq D$ represents a subset whose influence we want to scrub, termed the forgetting dataset. Accordingly, the complement of $D_f$, termed the remaining dataset, is $D_r = \{(\mathbf{x}_i, y_i)\}_{i=n_f+1}^n = D \backslash D_f$. The forgetting and remaining data can be represented by stacking the feature vectors and response variables as follows:

$$\mathbf{X}_f := [\mathbf{x}_1, \mathbf{x}_2, \ldots, \mathbf{x}_{n_f}] \in \mathbb{R}^{d \times n_f}, \quad \mathbf{y} := [y_1, y_2, \ldots, y_{n_f}]^\top \in \mathbb{R}^{n_f \times 1}$$

$$\mathbf{X}_r := [\mathbf{x}_{n_f+1}, \mathbf{x}_{n_f+2}, \ldots, \mathbf{x}_n] \in \mathbb{R}^{d \times (n-n_f)}, \quad \mathbf{y}^r := [y_{n_f+1}, y_{n_f+2}, \ldots, y_n]^\top \in \mathbb{R}^{(n-n_f) \times 1}$$

The overall dataset $\mathbf{X}$ and $\mathbf{y}$ are composed separately by concatenating $\mathbf{X}_r, \mathbf{X}_f$ and $\mathbf{y}_r, \mathbf{y}_f$.

**Learning Procedure** We consider the machine unlearning problem based on the fine-tuning method dividing the learning process into two distinct phases: Original Training and Fine-tuning (Unlearning). During the original training phase, we train a model on $n$ data points $\mathbf{X} \in \mathbb{R}^{d \times n}$ and obtain an original model $\mathbf{w}_o$ by optimizing $L(\mathbf{w}_o, D)$, where $L(\mathbf{w}, D)$ is defined as the mean-squared-error (MSE) loss: $L(\mathbf{w}, D) \triangleq \frac{1}{n} \|\mathbf{X}^\top \mathbf{w} - \mathbf{y}\|_2^2$. For the fine-tuning (unlearning) phase, we initialize with the original parameter $\mathbf{w}_o$ and proceed to retrain the model specifically on a subset of the remaining dataset $D_t \subseteq D_r$ by optimizing $L(\mathbf{w}_t, D_t)$, where $\mathbf{w}_t$ is the unlearn model by fine-tuning.

Since we work in the overparameterized regime, where $n<d$, each $\mathbf{w}$ can perfectly fit the dataset. We can express each solution $\mathbf{w}$ to the following optimization problem:

$$\text{Original training:} \quad \mathbf{w}_o = \underset{\mathbf{w}}{\arg\min}\|\mathbf{w}\|_2, \quad \text{s.t. } \mathbf{y} = \mathbf{X}^\top \mathbf{w} \tag{1}$$

$$\text{Unlearn via fine-tuning:} \quad \mathbf{w}_t = \underset{\mathbf{w}}{\arg\min}\|\mathbf{w} - \mathbf{w}_o\|_2, \quad \text{s.t. } \mathbf{y}_t = \mathbf{X}_t^\top \mathbf{w} \tag{2}$$

$$\text{Train from scratch:} \quad \mathbf{w}_g = \underset{\mathbf{w}}{\arg\min}\|\mathbf{w}\|_2, \quad \text{s.t. } \mathbf{y}_r = \mathbf{X}_r^\top \mathbf{w} \tag{3}$$

Our goal is to evaluate how the fine-tuning solution $\mathbf{w}_t$ differs from the golden model solution $\mathbf{w}_g$ which refers to retraining the model parameters from scratch over the remaining dataset $D_r$. Existing work has assessed machine unlearning performance from various perspectives (Graves et al., 2021; Becker & Liebig, 2022; Golatkar et al., 2020a; Song et al., 2019). In this paper, we focus particularly on the Unlearning Loss (UL) and Remaining Loss (RL), which refers to the model performance on the forgetting and remaining dataset respectively. These losses are defined as follows:

$$\text{RL:} \quad L(\mathbf{w}, D_r) = \frac{1}{n_r}\|\mathbf{X}_r^\top \mathbf{w} - \mathbf{y}_r\|_2^2, \quad \text{UL:} \quad L(\mathbf{w}, D_f) = \frac{1}{n_f}\|\mathbf{X}_f^\top \mathbf{w} - \mathbf{y}_f\|_2^2.$$

## 3 NAIVE FINE-TUNING METHODS FAIL TO UNLEARN

In empirical studies (Kurmanji et al., 2024; Warnecke et al., 2021; Golatkar et al., 2020a) and Table 1, it can be observed that fine-tuning may retain the utility of a model but struggles to forget. In this section, we revisit this phenomenon, aiming to explain why the vanilla fine-tuning method succeeds in retaining the model's utility on the remaining dataset but fails to forget the targeted data it was trained on.

### 3.1 DISTINCT FEATURES

To simplify our analysis, we first consider distinct features with the following assumption:

**Assumption 3.1.** The datasets $\mathbf{X}_f$ and $\mathbf{X}_r$ possess distinct non-zero features, while $\mathbf{w}_*$ embodies the coefficients applicable across all features.

**Remark 1.** The assumption implies that each of these datasets contains features that are unique to each dataset–there is no overlap in the features present in $\mathbf{X}_f$ and $\mathbf{X}_r$. Therefore, the remaining(forgetting) dataset matrix can be denoted as $\mathbf{X}_r^\top = [\mathbf{R}^\top, \mathbf{0}]$ and $\mathbf{X}_f^\top = [\mathbf{0}, \mathbf{F}^\top]$, where $\mathbf{R} \subseteq \mathbb{R}^{d_r \times (n-n_f)}$ and $\mathbf{F} \subseteq \mathbb{R}^{d_f \times n_f}$ correspond to the non-zero parts, $d_r$ and $d_f$ are the distinct feature numbers for remaining and forgetting data, respectively, and it satisfied that $d_r + d_f = d$. Additionally, it holds that $\mathbf{w}_* = \mathbf{w}_*^f + \mathbf{w}_*^r$, where $\mathbf{w}_*^f$ and $\mathbf{w}_*^r$ are the optimal solution such that $\mathbf{y}^f = \mathbf{X}_f^\top \mathbf{w}_*^f$ and $\mathbf{y}^r = \mathbf{X}_r^\top \mathbf{w}_*^r$. In an ideal scenario for classification tasks, each class possesses its own unique set of features that distinctly differentiates it from other classes. We later extended our analysis to overlapping features in Section 3.2.

**Theorem 3.2.** *Suppose a model is trained by the procedure 2 and 3 separately. Under the Assumption 3.1, it holds that*

- *RL: $L(\mathbf{w}_t, D_r) = 0$, UL : $L(\mathbf{w}_t, D_f) = 0$;*

- *RL: $L(\mathbf{w}_g, D_r) = 0$, UL: $L(\mathbf{w}_g, D_f) = \|\mathbf{w}_*^f\|_{\frac{1}{n_f}\mathbf{X}_f\mathbf{X}_f^\top}^2.$*

*Here, $\mathbf{w}_t$ refers to the unlearned model via fine-tuning, $\mathbf{w}_g$ refers to the model parameter retrained from scratch, RL and UL refer to the remaining loss on the remaining data and the unlearning loss on the forgetting data.*

Theorem 3.2 presents two interesting observations: 1) The fine-tuning model can perform perfectly on the remaining dataset, which indicates that the information from training data has been preserved from the original model, $\mathbf{w}_o$, to the unlearned model via fine-tuning, $\mathbf{w}_t$. 2) The loss of the fine-tuning model on the forgetting dataset consistently remains zero, which diverges from the

performance of the golden model. This suggests that the fine-tuning model is unable to forget the information it previously acquired from $\mathbf{w}_o$, which may be contradicted by catastrophic forgetting in continual learning (Parisi et al., 2019; Ding et al., 2024).

To illustrate the behavior of fine-tuning during the unlearning process more clearly, we consider the projective nature of learning. Firstly, the solution of Equation (2) can be represented as

$$\mathbf{w}_t = (\mathbf{I} - \mathbf{P}_t)\mathbf{w}_o + \mathbf{P}_t \mathbf{w}_*^r, \tag{4}$$

where $\mathbf{P}_t$ is the projection space of $\mathbf{X}_t$, the $\mathbf{I} - \mathbf{P}_t$ is the corresponding orthogonal space, and the $\mathbf{w}_o$ can be also represented $\mathbf{w}_o = \mathbf{P}\mathbf{w}_*$ with $\mathbf{P}$ being the projection space of $\mathbf{X}$. According to the property of projection Corollary B.1, multiplying any data matrix by a projection matrix preserves the components of the data that lie within the subspace defined by the projection. Moreover, under the distinct features assumption 3.1, it holds that

$$\mathbf{w}_t = \mathbf{P}\mathbf{w}_*^r + (\mathbf{P} - \mathbf{P}_t)\mathbf{w}_*^f. \tag{5}$$

Therefore, the unlearned model $\mathbf{w}_t$ from Equation (2) decomposed into two components for the unlearning process: the first part, $\mathbf{w}_*^r$, preserves the accuracy on the remaining data, while the second part, $\mathbf{w}_*^f$, also ensures accuracy on the forget data. However, the projection of $\mathbf{w}_*^f$ onto the fine-tuning space $\mathbf{P}_t$ has no effect, ultimately **resulting in the unlearned model $\mathbf{w}_t$ being exactly the same as the pretrained model $\mathbf{w}_o$.** The proof of Theorem 3.2 is provided in Appendix B.2.

### 3.2 Overlapping Features

In practical scenarios, training datasets often deviate from ideal classifications, introducing complexities such as overlapping features between subsets. This challenges assumptions of distinct feature sets across datasets. Therefore, we extend our previous analysis to address the presence of overlapped features. In the following, we begin by defining overlapped features.

**Assumption 3.3.** The datasets $\mathbf{X}_f$ and $\mathbf{X}_r$ possess $d_r$ overlapped features, while $\mathbf{w}_*$ embodies the coefficients applicable across all features.

**Remark 2.** Under Assumption 3.3, the dataset can be structured as follows: $\mathbf{X}_r^\top = [\mathbf{R}^\top, \mathbf{L}_1^\top, \mathbf{0}]$ and $\mathbf{X}_f^\top = [\mathbf{0}, \mathbf{L}_2^\top, \mathbf{F}^\top]$, where $\mathbf{R} \subseteq \mathbb{R}^{d_r \times n_r}$ and $\mathbf{F} \subseteq \mathbb{R}^{d_f \times n_f}$ represent the distinct features of the remaining and forgetting data, respectively. $\mathbf{L}_1 \subseteq \mathbb{R}^{d_{lap} \times n_r}$ and $\mathbf{L}_2 \subseteq \mathbb{R}^{d_{lap} \times n_f}$ denote the overlapped parts. Similarly to the Assumption 3.1, $d_r$ and $d_f$ are the distinct feature numbers for remaining and forgetting data, respectively, while the equation $d_r + d_{lap} + d_f = d$ holds. Additionally, the optimal solution can be decomposed into $\mathbf{w}_* = \mathbf{w}_*^f + \mathbf{w}_*^{lap} + \mathbf{w}_*^r$ such that $\mathbf{y}^f = \mathbf{X}_f^\top(\mathbf{w}_*^f + \mathbf{w}_*^{lap})$ and $\mathbf{y}^r = \mathbf{X}_r^\top(\mathbf{w}_*^r + \mathbf{w}_*^{lap})$.

**Theorem 3.4.** *Suppose a model is trained by the procedure 2 and 3 separately. Under the Assumption 3.3, it holds that*

- *RL: $L(\mathbf{w}_t, D_r) = 0$, UL: $L(\mathbf{w}_t, D_f) = 0$;*

- *RL: $L(\mathbf{w}_g, D_r) = 0$, UL: $L(\mathbf{w}_g, D_f) = \|\mathbf{P}_r \mathbf{w}_*^r + \mathbf{P}_r \mathbf{w}_*^{lap} - (\mathbf{w}_*^f + \mathbf{w}_*^{lap})\|_{\frac{1}{n_f}\mathbf{X}_f \mathbf{X}_f^\top}^2.$*

Theorem 3.4 shows that the previous conclusions remain valid under the assumptions of overlapping features, as the information from all training data, including forget data, is preserved from the pretrained model, $\mathbf{w}_o$, to the unlearned model through fine-tuning, $\mathbf{w}_t$. Consequently, the loss on both the remaining dataset and the forgetting dataset for the fine-tuning model is zero. Additionally, an interesting observation is that the number of overlapping features does not impact the unlearning accuracy of the fine-tuning model. The proof of Theorem 3.4 is provided in Appendix B.3.

Both Theorem 3.2 and Theorem 3.4 present similar findings regarding the performance of the unlearned model through fine-tuning. We run a synthetic experiment to validate these results (more experimental details in Appendix A). In Section 3.2 and Figure 1b, both distinct and overlapping feature assumptions demonstrate the same results: 1) The remaining loss of fine-tuning model $\mathbf{w}_t$ and golden model $\mathbf{w}_g$ is zero, indicating that the fine-tuning model performs equivalently to the golden model, successfully retaining the model's utility on the remaining dataset. 2) The unlearning loss of the fine-tuning model consistently remains at zero, differing from the golden model, suggesting that the fine-tuning model fails to forget the information obtained from the pretrained model. These empirical findings align well with our theoretical analysis.

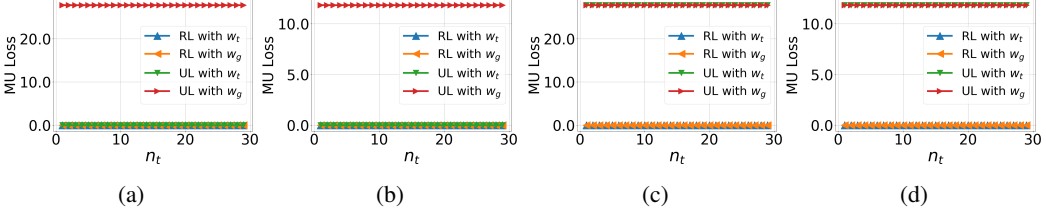

(a)                    (b)                    (c)                    (d)

Figure 1: Machine Unlearning Performance via (Regularized) Fine-tuning with (without) Overlapping Features. Section 3.2 and Figure 1b present the relationship between machine unlearning loss (i.e. RA, UA) and the number of fine-tuning data samples under distinct features and overlapping features assumptions, using naive FT method. In contrast, Figure 1c and Figure 1d show the same relationship using regularized fine-tuning methods, as discussed in Section 4.

# 4 ELIMINATING FORGETTING DATA FEATURES FROM PRE-TRAINED MODEL ENHANCES UNLEARNING

Compared to the golden model $\mathbf{w}_g = \mathbf{P}_r \mathbf{w}_*^r$, the unlearned model can be viewed as having an additional second term as

$$\mathbf{w}_t = \mathbf{P}\mathbf{w}_*^r + (\mathbf{P} - \mathbf{P}_t)\mathbf{w}_*^f.$$

This additional term $(\mathbf{P} - \mathbf{P}_t)\mathbf{w}_*^f$ represents the residual influence of the data intended to be forgotten on the unlearned model, contributing to the unlearning accuracy (UA) gap between $\mathbf{w}_t$ and the golden model $\mathbf{w}_g$. A natural approach to mitigate this gap might involve making the fine-tuning space converge toward the pretraining space-that is, aligning $\mathbf{P}_t$ with $\mathbf{P}_r$. However, this strategy is inefficient and contradictory, as it would lead to the optimal solution for the fine-tuning dataset becoming identical to that of the entire dataset, undermining the purpose and benefits of fine-tuning.

Inspired by the formulation of the unlearned model:

$$\mathbf{w}_t = (\mathbf{I} - \mathbf{P}_t)\mathbf{w}_o + \mathbf{P}_t\mathbf{w}_*^r.$$

To mitigate the UA gap between the fine-tuning model and the golden model, it becomes evident that the remaining portion of the pretrained model does not contribute to UA. Specifically, the components of the pretrained model $\mathbf{w}_o$ associated with the forgetting data $(\mathbf{w}_*^f)$ do not enhance performance on the remaining dataset $D_r$. Therefore, if we can eliminate the forgetting component—specifically by removing the $\mathbf{w}_*^f$ term from $\mathbf{w}_o$—the divergence can be addressed. In the following, we provide a formal description of this modification. Consider the same learning procedure Equation (1) to obtain the pretrained model $\mathbf{w}_o$. Prior to unlearning through fine-tuning, we modify $\mathbf{w}_o$ by removing components associated with the forgetting data. Specifically, we construct a modified model $\hat{\mathbf{w}}_o$ as follows:

1. **Distinct Features Scenario.** When the features of $D_r$ and $D_f$ are distinct, we construct $\hat{\mathbf{w}}_o$ by retaining only the components corresponding to $D_r$ and setting the rest to zero. Formally, we define $\hat{\mathbf{w}}_o$ as $\hat{\mathbf{w}}_o'[0 : d_r] = \mathbf{w}_o[0 : d_r]$ or equivalently can be understood as:

$$\hat{\mathbf{w}}_o[i] = \begin{cases} \mathbf{w}_o[i], & \text{if } i \in \text{features of } D_r, \\ 0, & \text{otherwise.} \end{cases}$$

2. **Overlapping Features Scenario.** When features overlap across $D_r$ and $D_f$, we consider two cases:

   - **Option A** (Retaining Overlapping Features): We retain the overlapping features between $D_r$ and $D_f$, which can be expressed as $\hat{\mathbf{w}}_o[0 : d_r + d_{lap}] = \mathbf{w}_o[0 : d_r + d_{lap}]$ or equivalently

   $$\hat{\mathbf{w}}_o[i] = \begin{cases} \mathbf{w}_o[i], & \text{if } i \in \text{features of } D_r \cup \text{overlapping features}, \\ 0, & \text{otherwise.} \end{cases}$$

   - **Option B** (Discarding Overlapping Features): We discard the overlapping features, which can be expressed as $\hat{\mathbf{w}}_o'[0 : d_r] = \mathbf{w}_o[0 : d_r]$ or equivalently

   $$\hat{\mathbf{w}}_o'[i] = \begin{cases} \mathbf{w}_o[i], & \text{if } i \in \text{features of } D_r, \\ 0, & \text{otherwise.} \end{cases}$$

**Theorem 4.1.** *Let $\mathbf{w}_o$ be a pretrained model obtained the overall dataset $D = D_r \cup D_f$. Before performing unlearning (fine-tuning), we modify $\mathbf{w}_o$ to remove the components associated with $D_f$ as described above. Then, using the modified models $\hat{\mathbf{w}}_o$ ($\hat{\mathbf{w}}_o'$) in the unlearning process, we have:*

1. ***Distinct Features Scenario** (Under the Assumption 3.1), we have:*
   *RL: $L(\hat{\mathbf{w}}_t, D_r) = 0$; UL: $L(\hat{\mathbf{w}}_t, D_f) = \|\mathbf{w}_*^f\|_{\frac{1}{n_f}\mathbf{X}_f\mathbf{X}_f^\top}^2$,*

2. ***Overlapping Features Scenario** (Under Assumption 3.3), we have:*

   - ***Option A** (Retaining Overlapping Features):*
     *RL: $L(\hat{\mathbf{w}}_t, D_r) = 0$;*
     *UL: $L(\hat{\mathbf{w}}_t, D_f) = \|\mathbf{P}\mathbf{w}_*^r + \mathbf{P}\mathbf{w}_*^{lap} - (\mathbf{w}_*^f + \mathbf{w}_*^{lap})\|_{\frac{1}{n_f}\mathbf{X}_f\mathbf{X}_f^\top}^2.$*

   - ***Option B** (Discarding Overlapping Features):*
     *RL: $L(\hat{\mathbf{w}}_t', D_r) = \|(\mathbf{I} - \mathbf{P}_t)\mathbf{w}_*^{lap}\|_{\frac{1}{n_r}\mathbf{X}_r\mathbf{X}_r^\top}^2$*
     *UL: $L(\hat{\mathbf{w}}_t', D_f) = \|\mathbf{P}\mathbf{w}_*^r + \mathbf{P}_t\mathbf{w}_*^{lap} - (\mathbf{w}_*^f + \mathbf{w}_*^{lap})\|_{\frac{1}{n_f}\mathbf{X}_f\mathbf{X}_f^\top}^2.$*

According to Theorem 4.1, under the distinct features assumption, the regularized unlearned model achieves the same remaining and unlearning loss as the golden model. Furthermore, when considering overlapping features, if the overlapped component from the pretrained model is retained, the remaining loss remains zero, as with the golden model, while the unlearning loss differs only in the projection component. This difference can be considered negligible when applied to $\mathbf{w}_*^r$ and $\mathbf{w}_*^{lap}$ due to the model assumption. Figure 1c and Figure 1d verify our theoretical conclusions. However, if the overlapped component is discarded from the pretrained model, the remaining loss is no longer zero, and there is a small change to the unlearning loss that can be overlooked.

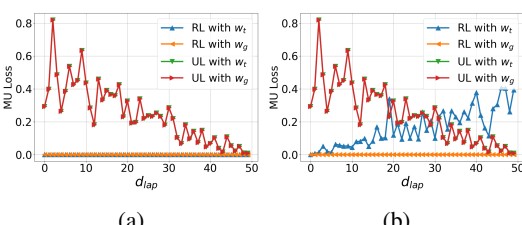

(a)  (b)

Figure 2: Comparison of Machine Unlearning Loss with and without Overlapping Features. Figure 2a retains overlapping features from the pretrained model, showing the matching performance between regularized $\mathbf{w}_t$ model and golden model $\mathbf{w}_g$; Figure 2b discards the overlapping features, showing a decline in retaining accuracy.

These findings offer several insights into the design of machine unlearning algorithms: **1) Regularization on the pretrained model can significantly improve unlearning accuracy while preserving the retaining accuracy**. If we can identify the component of the pretrained model related to the forgetting data, applying regularization to this component can further enhance UA. Our theorem also explains recent related works, such as Liu et al. (2024); Fan et al. (2023), which apply a mask to the pretrained model either randomly or by regularizing the weights associated with the forgetting data to provide better unlearn performance. These methods share the same underlying principle discussed here. **2) When considering overlapping features, retaining them does not substantially affect unlearning accuracy, but discarding them compromises the retaining accuracy**. As shown in Theorem 4.1, the remaining loss can not retain zero unless the remaining data $\mathbf{X}_r$ can be fully represented by the fine-tuning space, meaning $\mathbf{P}_t\mathbf{X}_r = \mathbf{X}_r$. Additionally, as the number of overlapping features increases, the impact on both remaining and unlearning loss becomes more significant. Discarding too many overlapping components can lead to instability in the retaining accuracy, as the model loses essential information needed to represent $D_r$, which in turn causes the remaining loss to increase. Figure 1 and Figure 2 validate our theoretical findings. The proof of Theorem 4.1 is provided in Appendix B.4.

## 5 REVISITING DISCRIMINATIVE REGULARIZATION

In Section 4 we show that once the components of the pretrained model related to the forgetting data are identified and removed, unlearning accuracy can be significantly improved. However, in practice,

the training dataset and model are often not well-structured (Assumption 3.1 and Assumption 3.3 may not hold). In such cases, as motivated by Equation (5), we know that the fine-tuning space may fail to unlearn from the forgetting data, allowing all information from the pretrained model to be retained. This raises the question: *what happens if the fine-tuning space learns incorrect or faulty information about the forgetting data?* A recent study Fan et al. (2023) addresses this issue by introducing a regularized constraint in the fine-tuning unlearning process. This approach ensures that fine-tuning not only preserves the utility of the model on the remaining data but also effectively forgets the target data. Specifically, it achieves saliency-based unlearning by minimizing the following optimization problem:

$$\text{CE-FT:} \quad \min_{\mathbf{w}_t} \underbrace{\mathcal{L}_{\text{CE}}(\mathbf{w}_t{}^1; \mathbf{X}_f, \mathbf{Y}'_f)}_{\text{for unlearning accuracy}} + \underbrace{\alpha \mathcal{L}_{\text{CE}}(\mathbf{w}_t; \mathbf{X}, \mathbf{Y})}_{\text{regularization for retaining accuracy}} \quad . \tag{6}$$

where the first term is the cross entropy loss function used to measure the model's accuracy on the forget dataset, which is intentionally mislabeled Golatkar et al. (2020a). This term acts as a penalty, encouraging the model to reduce its ability to accurately predict the target data, thereby facilitating the unlearning process. $\alpha$ is a regularization parameter that balances the trade-off between maintaining accuracy on the retaining data and forgetting the target data. The second term corresponds to the cross entropy loss function applied to the retaining data.

Notably, the regularization parameter is typically constrained to the range $(0, 1]$. However, based on our previous analysis, **we favor the principle that regularization should prioritize retaining accuracy over unlearning accuracy** (since retaining overlapping features does not significantly impact UA, but discarding them compromises RA). Therefore, we will explore the impact of switching the regularization focus in Equation (6), which we refer to as Inverse CE (ICE).

Additionally, it can be observed that Equation (6) relies exclusively on cross-entropy loss for both retaining and unlearning accuracy. This approach emphasizes the discrepancy between the true labels and predicted probabilities, focusing on the correct class while penalizing incorrect predictions. In our paper, we hope to ensure that the fine-tuning process learns an incorrect distribu-

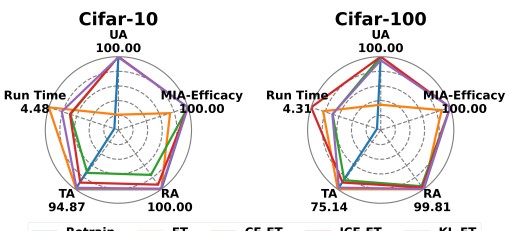

Figure 3: Performance comparison of fine-tuning methods on CIFAR-10 and CIFAR-100 datasets using five metrics: Unlearning Accuracy (UA), MIA-Efficacy, Retaining Accuracy (RA), Test Accuracy (TA), and Run Time. Each metric is normalized to the range [0, 1] based on the best result across all unlearning methods for ease of visualization, with the actual best value provided alongside each metric.

tion for the forgetting dataset. To achieve this and provide a comprehensive discussion, we also include KL-Divergence as an additional loss function. Specifically, we define the following:

$$\text{KL-FT:} \quad \min_{\mathbf{w}_t} \underbrace{\mathcal{L}_{\text{CE}}(\mathbf{w}_t; \mathbf{X}, \mathbf{Y})}_{\text{for retaining accuracy}} + \underbrace{\alpha \mathcal{L}_{\text{KL}}(\mathbf{w}_t; \mathbf{X}_f, \mathbf{Y}'_f)}_{\text{regularization for unlearning accuracy}} \quad . \tag{7}$$

# 6 EXPERIMENT

In this section, we verify our theoretical insights by evaluating the effectiveness of the regularization-based FT machine unlearning methods through numerical experiments.

## 6.1 EXPERIMENT SETUPS

**Datasets and Models.** The baseline method is the naive fine-tuning approach (Golatkar et al., 2020a; Warnecke et al., 2021) implemented on ResNet-18 (He et al., 2016) and we also include the golden retrained model for comparison. Our experiments will focus on image classification using the CIFAR-10 (Krizhevsky et al., 2009), CIFAR-100 (Krizhevsky et al., 2009), and SVHN (Netzer et al., 2011) datasets. More details on the experimental setup will be provided in Appendix A.

---

[1]To maintain consistency in the optimization problem, we do not incorporate the use of a mask in Fan et al. (2023), instead focusing solely on regularization-based FT methods. We will leave it as a future work.

**Evaluation Metrics.** We follow the existing work to assess machine unlearning performance from different aspects (Golatkar et al., 2020a; Graves et al., 2021; Thudi et al., 2022; Liu et al., 2024; Sharma et al., 2024; Zhu et al., 2024). Specifically, we focus on the following evaluation metrics:

- Unlearning accuracy (UA): We define $\text{UA}(\mathbf{w}_t) = 1 - \text{Acc}_{D_f}(\mathbf{w}_t)$ as Liu et al. (2024), measuring how effectively the model has forgotten the targeted data. Here $\text{Acc}_{D_f}(\mathbf{w}_t)$ is the accuracy of the unlearned model on the forgetting dataset.

- Membership inference attack (MIA) on $D_f$ (MIA-Efficacy): The efficacy of MIA on the forget dataset, which assesses whether the model still retains any identifiable information about the forgetting data.

- Retaining accuracy (RA): The accuracy of the model on the remaining dataset $D_r$ after unlearning, measuring how well the model retains its performance from pretrained model.

- Testing accuracy (TA): The accuracy of the model on the independent test dataset, indicating its generalization ability after unlearning.

- Run-time efficiency (RTE): RTE evaluates the computational efficiency of the unlearning process, including the run-time cost taken to execute the unlearning procedure.

Note that a smaller performance gap between the unlearned model and the golden retrained model indicates the better performance of approximate unlearning.

## 6.2 Experiment Results

Table 2: **Cifar-10, Cifar-100, SVHN Class-wise Forgetting.** This table presents the performance of different unlearning methods, including FT, Sparse-FT, CE-FT, ICE-FT, and KL-FT, across CIFAR-10, CIFAR-100, and SVHN datasets. Results are reported as mean ± standard deviation over five independent trials. A performance gap is computed against the retrain method, showing how each approach performs relative to the golden retraining method.

| | Cifar-10 Class-wise Forgetting | | | | |
| **Methods** | **UA** | **MIA-Efficacy** | **RA** | **TA** | **Run Time** |
| --- | --- | --- | --- | --- | --- |
| Retrain | $100.00_{\pm 0.00}$ | $100.00_{\pm 0.00}$ | $100.00_{\pm 0.00}$ | $94.87_{\pm 0.14}$ | 77.00 |
| FT | $20.89_{\pm 4.12}(79.11)$ | $74.32_{\pm 11.90}(25.68)$ | $99.76_{\pm 0.12}(0.24)$ | $93.73_{\pm 0.22}(1.14)$ | 4.48 |
| CE-FT | $100.00_{\pm 0.00}(0.00)$ | $100.00_{\pm 0.00}(0.00)$ | $75.76_{\pm 5.03}(24.24)$ | $68.67_{\pm 5.11}(26.20)$ | 6.49 |
| ICE-FT | $100.00_{\pm 0.00}(0.00)$ | $100.00_{\pm 0.00}(0.00)$ | $92.22_{\pm 0.72}(7.78)$ | $84.22_{\pm 1.12}(10.65)$ | 6.43 |
| KL-FT | $99.17_{\pm 0.29}(0.83)$ | $100.00_{\pm 0.00}(0.00)$ | $99.06_{\pm 0.51}(0.94)$ | $92.54_{\pm 0.67}(2.33)$ | 5.50 |

| | Cifar-100 Class-wise Forgetting | | | | |
| **Methods** | **UA** | **MIA-Efficacy** | **RA** | **TA** | **Run Time** |
| --- | --- | --- | --- | --- | --- |
| Retrain | $100.00_{\pm 0.00}$ | $100.00_{\pm 0.00}$ | $99.81_{\pm 0.06}$ | $75.14_{\pm 0.12}$ | 81.00 |
| FT | $34.44_{\pm 19.89}(65.56)$ | $87.64_{\pm 10.31}(12.36)$ | $99.79_{\pm 0.10}(0.21)$ | $75.07_{\pm 0.56}(0.07)$ | 5.20 |
| CE-FT | $99.87_{\pm 0.13}(0.13)$ | $99.95_{\pm 0.05}(0.05)$ | $94.71_{\pm 1.01}(5.10)$ | $63.64_{\pm 0.93}(11.50)$ | 6.23 |
| ICE-FT | $100.00_{\pm 0.00}(0.00)$ | $100.00_{\pm 0.00}(0.00)$ | $96.66_{\pm 1.28}(3.15)$ | $66.65_{\pm 2.08}(8.49)$ | 4.31 |
| KL-FT | $95.20_{\pm 2.31}(4.80)$ | $100.00_{\pm 0.00}(0.00)$ | $99.26_{\pm 0.16}(0.55)$ | $73.11_{\pm 0.42}(2.03)$ | 6.20 |

| | SVHN Class-wise Forgetting | | | | |
| **Methods** | **UA** | **MIA-Efficacy** | **RA** | **TA** | **Run Time** |
| --- | --- | --- | --- | --- | --- |
| Retrain | $100.00_{\pm 0.00}$ | $100.00_{\pm 0.00}$ | $100.00_{\pm 0.00}$ | $88.20_{\pm 0.75}$ | 72.00 |
| FT | $17.47_{\pm 7.29}(82.53)$ | $99.95_{\pm 0.05}(0.05)$ | $100_{\pm 0.00}(0.00)$ | $93.55_{\pm 0.63}(5.35)$ | 5.01 |
| CE-FT | $100_{\pm 0.00}(0.00)$ | $100_{\pm 0.00}(0.00)$ | $97.27_{\pm 2.70}(3.73)$ | $79.77_{\pm 3.51}(10.87)$ | 5.48 |
| ICE-FT | $100.00_{\pm 0.00}(0.00)$ | $100.00_{\pm 0.00}(0.00)$ | $99.99_{\pm 0.01}(0.01)$ | $85.56_{\pm 0.32}(2.65)$ | 4.48 |
| KL-FT | $97.24_{\pm 0.90}(2.76)$ | $100.00_{\pm 0.00}(0.00)$ | $99.95_{\pm 0.05}(0.05)$ | $87.54_{\pm 0.15}(0.66)$ | 5.23 |

**Performance Comparison Among Regularization-Based Fine-Tuning Methods.** In Table 2, we explore the impact of different regularization terms on the performance of various FT-based methods. It is evident that the regularization term consistently enhances the model's unlearning accuracy. Specifically, in the CIFAR-10 experiments, all regularization-based methods show a significant improvement in UA compared to the baseline FT (20.89%). Both ICE-FT and CE-FT achieve perfect performance in UA and MIA-Efficacy, showing no gap with the golden model. However, it is also noticeable that their RA and TA, especially for the CE-FT method, decrease dramatically, indicating that the improvement in UA comes at the expense of reduced RA and TA. Notably, our KL-FT method achieves the most comparable RA of 99.06% and TA of 92.54% relative to the baseline Retrain (100.00% and 94.87%) and FT (99.76% and 93.73%), without the extreme RA decline seen

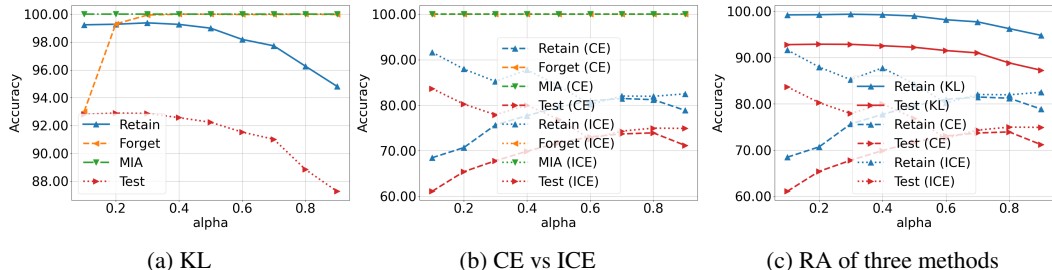

(a) KL      (b) CE vs ICE      (c) RA of three methods

Figure 4: This figure shows the impact of varying the regularization parameter $\alpha$ on the accuracy metrics for KL-FT, CE-FT, and ICE-FT. Figure 4a illustrates the sensitivity of the KL method across various evaluation metrics within the range of 0 to 1. Figure 4b shows the impact of different regularization focuses by comparing the performance of CE-FT and ICE-FT. Figure 4c presents the RA behavior for all three methods (KL, CE, and ICE-FT).

in other methods. Similarly, in the CIFAR-100 experiments, KL-FT achieves the best RA of $99.26\%$ and TA of $73.11\%$, outperforming all other methods, with only a minor decline in UA. These highlight the effectiveness of KL-FT in balancing unlearning and retaining accuracy. For the SVHN dataset, the FT method shows even more challenges in forgetting classes, with a UA of $17.47\%$. In contrast, all regularization-based methods, including CE-FT, ICE-FT, and KL-FT, achieve perfect UA, signifying complete forgetting. ICE-FT stands out with the best TA of $85.56\%$, closely followed by KL-FT with a TA of $87.54\%$.

ICE-FT builds upon the CE-FT by adjusting the focus of regularization. Instead of prioritizing forgetting during the unlearning process, ICE-FT balances the two, placing more emphasis on RA. This change allows ICE-FT to achieve perfect unlearning (UA of $100.00\%$) while maintaining significantly higher RA and TA compared to CE-FT across all datasets. These results align with our previous theoretical analysis, which suggests that focusing on retention does not significantly impact UA, but shifting the focus away from retention compromises RA.

**Sensitivity of regularization parameter $\alpha$.** The regularization parameter $\alpha$ is a crucial hyperparameter in regularization-based FT method. To demonstrate its effect, we conduct numerical experiments on the CIFAR-10 dataset, showing how the regularization parameter impacts the unlearning performance of various regularization-based FT methods. We first examine the sensitivity of the KL method across various evaluation metrics within the range of $(0, 1]$. As shown in Figure 4a, as $\alpha$ increases, the RA gradually declines from near-perfect performance ( $\sim 100\%$ ) down to $\sim 94\%$ at $\alpha = 0.8$, indicating that stronger regularization negatively affects the retention of information. Meanwhile, the Test Accuracy follows the same downward trend as the Retaining Accuracy, dropping from $\sim 94\%$ to $\sim 88\%$. Additionally, we explore how different regularization focuses impact unlearning performance. In Figure 4b, we observe that the RA of CE-FT decreases gradually with increasing $\alpha$, and UA rises slightly before stabilizing, the MIA-Efficacy stays consistently high. In contrast, the ICE-FT method prioritizes retaining accuracy on the remaining data, resulting in a higher Retaining Accuracy than CE-FT across all $\alpha$ values. The UA slightly decreases but remains competitive, while MIA-Efficacy and Test Accuracy follow similar trends to those seen in CE-FT. This suggests that ICE-FT achieves a better balance between unlearning and accuracy retention than standard CE-FT, which aligns with our previous analysis. Finally, we illustrate the RA and TA behavior for all three methods in Figure 4c.

## 7   Conclusion

In conclusion, we present the first theoretical analysis of fine-tuning methods for machine unlearning within a linear regression framework. Our analysis, covering two scenarios—distinct and overlapping feature sets—demonstrates that while fine-tuning can achieve optimal retaining accuracy (RA), it fails to fully unlearn the forgetting dataset. Our analysis on the failure of naive fine-tuning methods stems from the pretrained model's retention of forgetting data, and we propose a theoretical approach to mitigate this issue. By revisiting and redesigning the discriminative regularization term, we prioritize retaining accuracy while effectively balancing it with unlearning accuracy. Experimental results on both synthetic and real-world datasets validate our theoretical insights, demonstrating that our redesigned regularization approach significantly enhances unlearning performance without sacrificing retention.

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

# A EXPERIMENTAL DETAILS

## A.1 VERIFICATION VIA SIMULATION

To empirically validate the theoretical findings presented in Theorem 3.2, Theorem 3.4 and Theorem 4.1 regarding the performance of unlearned models through fine-tuning, we first conducted a series of synthetic experiments.

**Data Generation.** We constructed two data matrices, $\mathbf{X}_r$ and $\mathbf{X}_f$, representing the remaining data and the forgetting data, respectively. The remaining data matrix, $\mathbf{X}_r^\top$, was structured as $[\mathbf{R}^\top, \mathbf{L}_1^\top, \mathbf{0}]$, and the forgetting data matrix, $\mathbf{X}_f^\top$, as $[\mathbf{0}, \mathbf{L}_2^\top, \mathbf{F}^\top]$, where $\mathbf{L}_1^\top = \mathbf{L}_2^\top = \mathbf{0}$ to enforce the non-overlapping case. Here, $\mathbf{R}^\top$ and $\mathbf{F}^\top$ are random matrices corresponding to different feature sets, and the zeros represent the distinct features across the datasets. We set the total number of data points to $n = 40$ and the total number of features to $d = 40$. The remaining data consisted of $n_r = 30$ samples with $d_r = 20$ features, while the forgetting data comprised $n_f = 10$ samples with $d_f = d - d_r = 20$ features. To simulate a controlled environment, we fixed the number of overlapping features to $d_{lap} = 0$ and $d_{lap} = 8$ for non-overlapping case and overlapping case, respectively.

**Label Generation.** We generated the true coefficient vector $\mathbf{w}_* \in \mathbb{R}^d$ by sampling from a standard normal distribution. The labels were created using a linear regression model without added noise:$\mathbf{y} = \mathbf{X}^\top \mathbf{w}_*$. The labels were partitioned into $\mathbf{y}_r$ and $\mathbf{y}_f$, corresponding to the remaining and forgetting data, respectively.

**Model Training.** To compare the effects of fine-tuning, we considered two models: the fine-tuning model $\mathbf{w}_t$ and the golden model $\mathbf{w}_g$. Specifically, $\mathbf{w}_t$ was obtained by fine-tuning on a subset of the remaining data, denoted as $\mathbf{X}_t$, which consisted of the first $n_t$ data points from $\mathbf{X}_r$. The value of $n_t$ varied from 1 to $n_r - 1$ to study the impact of the fine-tuning data size. The initial model $\mathbf{w}_o$ was derived from the entire dataset $\mathbf{X}$ and calculated by the Equation (1). $\mathbf{w}_g$ was trained from scratch on the entire remaining data $\mathbf{X}_r$ and computed by solving Equation (2). If considering the regularization case in synthetic data, the regularized pretrained model will be constructed by zeroing out the coefficients corresponding to the forgetting data features with (without) overlapping features.

**Evaluation Metrics.** The performance of the models was assessed using the Mean Squared Error (MSE) on both the remaining and forgetting data:

- Remaining Data Loss (RA Loss):$\mathrm{MSE}_{\mathrm{RA}}(\mathbf{w}) = \frac{1}{n_r}\|\mathbf{X}_r\mathbf{w} - \mathbf{y}_r\|^2$

- Unlearning Data Loss (UA Loss):$\mathrm{MSE}_{\mathrm{UA}}(\mathbf{w}) = \frac{1}{n_f}\|\mathbf{X}_f\mathbf{w} - \mathbf{y}_f\|^2$.

**Experimental Results** Figure 1c and Figure 1d illustrate that the regularized fine-tuning method discussed in Section 4 can significantly improve unlearning accuracy while preserving the retaining accuracy. Specifically, both the remaining loss and unlearning loss of $\hat{\mathbf{w}}_t$ perfectly match those of the golden model under both distinct and overlapping feature scenarios. Additionally, Figure 2 present comparisons of machine unlearning loss for different approaches to handling overlapping features: Figure 2a retains overlapping features from the pretrained model, demonstrating matching performance between the regularized $\mathbf{w}_t$ model and golden model $\mathbf{w}_g$; whereas Figure 2b discards the overlapping features, resulting in a decline in retaining accuracy. These empirical results align well with our theoretical findings.

## A.2 ADDITIONAL REAL-WORLD EXPERIMENTS

**Unlearning Setup** The unlearning setup centers on the FT-based procedure. During training, the model is updated using the remaining dataset, while Kullback–Leibler divergence/Cross-entropy loss regularization is applied to the forget dataset to enforce unlearning. The corresponding regularization modifies the model's predictions by encouraging it to generate incorrect outputs for the forgetting data. Specifically, KL divergence is computed between the model's output and shifted incorrect labels, ensuring the model no longer retains knowledge of the forgetting data. Additionally, cross-entropy loss between the model's output and the incorrect labels further supports the unlearning process. Throughout training, the optimizer updates the model based on the combined loss. Our

experiments focus on class-wise forgetting, and we run the process 5 times, reporting the mean and standard deviation of the performances.

We summarize the datasets and model configurations in Tab. 3.

Table 3: Dataset and model setups.

| Dataset | CIFAR-10 | SVHN | CIFAR-100 |
|---|---|---|---|
| Settings | ResNet-18 | ResNet-18 | ResNet-18 |
| Batch Size | 256 | 256 | 256 |

Table 4: Comparison of CE and ICE results across various $\alpha$ values on Cifar-10 dataset.

| $\alpha$ | CE-FT | | | | ICE-FT | | | |
|---|---|---|---|---|---|---|---|---|
| | UA | MIA | RA | Test | UA | MIA | RA | Test |
| 0.1 | 100.00 | 100.00 | 68.48 | 61.11 | 100.00 | 100.00 | 91.66 | 83.67 |
| 0.2 | 100.00 | 100.00 | 70.7 | 65.41 | 100.00 | 100.00 | 87.97 | 80.27 |
| 0.3 | 100.00 | 100.00 | 75.67 | 67.81 | 100.00 | 100.00 | 85.24 | 77.92 |
| 0.4 | 100.00 | 100.00 | 77.73 | 69.91 | 100.00 | 100.00 | 87.79 | 80.02 |
| 0.5 | 100.00 | 100.00 | 79.6 | 71.73 | 100.00 | 100.00 | 84.31 | 76.89 |
| 0.6 | 100.00 | 100.00 | 80.97 | 73.06 | 100.00 | 100.00 | 80.18 | 72.56 |
| 0.7 | 100.00 | 100.00 | 81.49 | 73.71 | 100.00 | 100.00 | 82.03 | 74.33 |
| 0.8 | 100.00 | 100.00 | 81.23 | 73.94 | 100.00 | 100.00 | 81.97 | 74.98 |
| 0.9 | 100.00 | 100.00 | 78.89 | 71.2 | 100.00 | 100.00 | 82.51 | 74.92 |

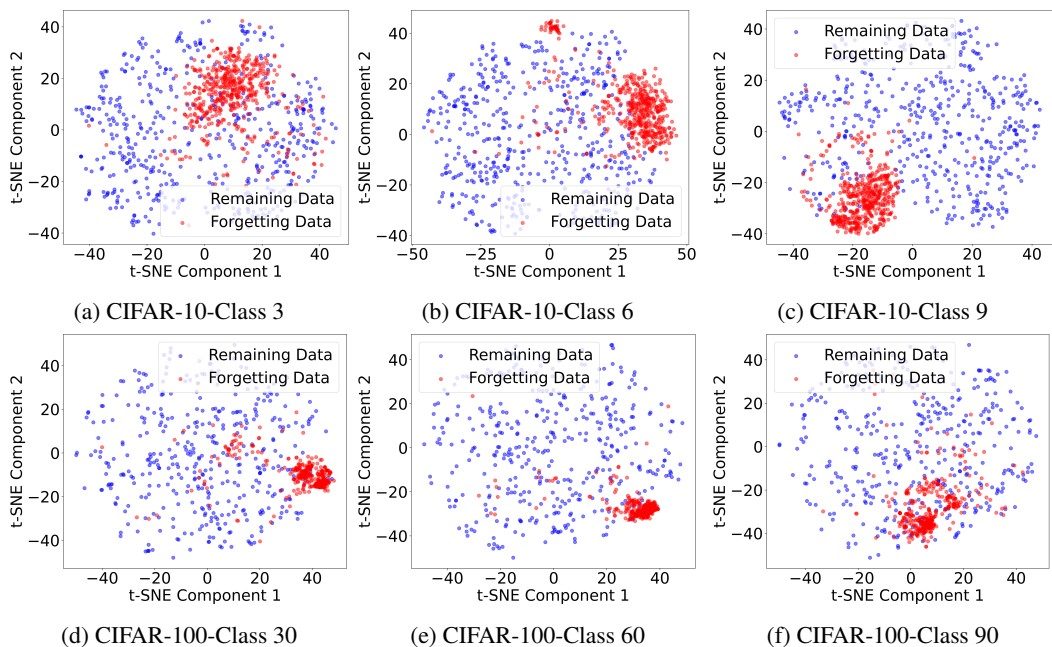

(a) CIFAR-10-Class 3  (b) CIFAR-10-Class 6  (c) CIFAR-10-Class 9

(d) CIFAR-100-Class 30  (e) CIFAR-100-Class 60  (f) CIFAR-100-Class 90

Figure 5: Visualization of Remaining Data and Forgetting Data Features Across Various Dataset. Figures 5a-5c focus on classes 3, 6, and 9 in CIFAR-10 and Figures 5d-5f focus on classes 30, 60, and 90 in CIFAR-100.

**Visualization of Remaining Data and Forgetting Data Features.** The visualization in Figure 5 uses t-SNE to project feature representations of the forgetting and remaining data in CIFAR-10 and

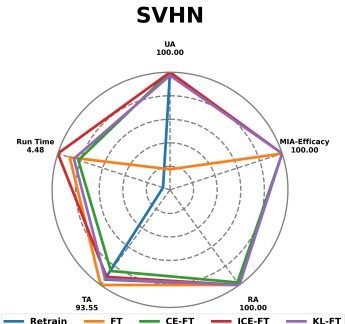

Figure 6: Performance comparison of fine-tuning methods on the SVHN dataset using five metrics.

CIFAR-100 datasets. The red points correspond to forgetting data, and the blue points represent remaining data. This visualization aims to demonstrate that, in class-wise datasets, the unlearning task for a specific class may involve distinct features. In such cases, naive fine-tuning (FT) methods tend to contribute less towards forgetting the class and focus more on retaining features from the pretrained model.

# B    PROOFS

## B.1    USEFUL PROPERTIES

Before presenting the detailed proofs of the theorems, we first introduce several useful properties of the projection matrix and the minimum norm solution.

**Property 1** (Projection properties)**.** Let $\mathbf{P}$ be a projection operator that projects onto a subspace $\mathbf{X} \subseteq \mathbb{R}^{d \times n}$. Then, $\mathbf{P}$ holds the following properties:

1. Symmetric: $\mathbf{P} = \mathbf{P}^\top$;

2. Idempotent: $\mathbf{P}^2 = \mathbf{P}$;

3. $\mathbf{I} - \mathbf{P}$ is also a projection operator, projecting onto the subspace orthogonal to $\mathbf{X}$. Therefore, $(\mathbf{I} - \mathbf{P})\mathbf{P} = \mathbf{0}$;

4. Let $\mathbf{v} \in \mathbb{R}^d$ be an arbitrary vector, it holds that $\|(\mathbf{I} - \mathbf{P})\mathbf{v}\|^2 = \mathbf{v}^\top (\mathbf{I} - \mathbf{P})^2 \mathbf{v} = \mathbf{v}^\top (\mathbf{I} - \mathbf{P})\mathbf{v} = \|\mathbf{v}\|^2 - \|\mathbf{P}\mathbf{v}\|^2$;

5. Contraction: $\|\mathbf{P}\mathbf{v}\| \leq \|\mathbf{v}\|$, holding in equality if and only if $\mathbf{P}\mathbf{v} = \mathbf{v}$.

*Proof.* See (Zarantonello, 1971) for the proofs and for more properties. ☐

**Corollary B.1** (Projection Matrix properties)**.** *Let* $\mathbf{P} = \mathbf{X}(\mathbf{X}^\top \mathbf{X})^{-1}\mathbf{X}^\top, \mathbf{P}_r, \mathbf{P}_f, \mathbf{P}_t$ *be the corresponding projection operator for* $\mathbf{X}, \mathbf{X}_r, \mathbf{X}_f, \mathbf{X}_t$ *respectively. Under Assumption 3.1, the remaining(forgetting) dataset matrix can be denoted as* $\mathbf{X}_r = \begin{bmatrix} \mathbf{R} \\ \mathbf{0} \end{bmatrix}$ *and* $\mathbf{X}_f = \begin{bmatrix} \mathbf{0} \\ \mathbf{F} \end{bmatrix}$, *where* $\mathbf{R} \subseteq \mathbb{R}^{d_r \times (n - n_f)}$ *and* $\mathbf{F} \subseteq \mathbb{R}^{d_f \times n_f}$ *correspond to the non-zero parts. Then, it holds that:*

*1.* $\mathbf{P} = \begin{bmatrix} \mathbf{R}(\mathbf{R}^\top \mathbf{R})^{-1}\mathbf{R}^\top & \mathbf{0} \\ \mathbf{0} & \mathbf{F}(\mathbf{F}^\top \mathbf{F})^{-1}\mathbf{F}^\top \end{bmatrix} = \mathbf{P}_r + \mathbf{P}_f;$

*2.* $\mathbf{P}_r = \begin{bmatrix} \mathbf{R}(\mathbf{R}^\top \mathbf{R})^{-1}\mathbf{R}^\top & \mathbf{0} \\ \mathbf{0} & \mathbf{0} \end{bmatrix}$ *and* $\mathbf{P}_f = \begin{bmatrix} \mathbf{0} & \mathbf{0} \\ \mathbf{0} & \mathbf{F}(\mathbf{F}^\top \mathbf{F})^{-1}\mathbf{F}^\top \end{bmatrix};$

*3.* $\mathbf{X}(\mathbf{I} - \mathbf{P}) = (\mathbf{I} - \mathbf{P})\mathbf{X} = 0$, *and the conclusion also holds for* $\mathbf{P}_r, \mathbf{P}_f, \mathbf{P}_t$ *with* $\mathbf{X}_r, \mathbf{X}_f, \mathbf{X}_t$ *respectively;*

4. *For any matrix $\mathbf{A}$ that is a submatrix of $\mathbf{X}$, it holds that $\mathbf{A} = \mathbf{PA}$, where $\mathbf{P}$ is the projection space of $\mathbf{X}$. Moreover, if $\mathbf{P}_A$ is the projection space of $\mathbf{A}$, it holds that $\mathbf{PP}_A = \mathbf{P}_A$, i.e. $\mathbf{X}_r\mathbf{P} = \mathbf{X}_r$, $\mathbf{X}_f\mathbf{P} = \mathbf{X}_f$, $\mathbf{X}_r\mathbf{P}_f = \mathbf{X}_f\mathbf{P}_r = 0$.*

**Proof of Corollary B.1.** Firstly, based on the data composition, the overall dataset holds $\mathbf{X} = \begin{bmatrix} \mathbf{R} & \mathbf{0} \\ \mathbf{0} & \mathbf{F} \end{bmatrix}$. Therefore, it follows:

$$\mathbf{P} = \mathbf{X}(\mathbf{X}^\top\mathbf{X})^{-1}\mathbf{X}^\top = \begin{bmatrix} \mathbf{R} & \mathbf{0} \\ \mathbf{0} & \mathbf{F} \end{bmatrix}(\begin{bmatrix} \mathbf{R}^\top & \mathbf{0} \\ \mathbf{0} & \mathbf{F}^\top \end{bmatrix}\begin{bmatrix} \mathbf{R} & \mathbf{0} \\ \mathbf{0} & \mathbf{F} \end{bmatrix})^{-1}\begin{bmatrix} \mathbf{R}^\top & \mathbf{0} \\ \mathbf{0} & \mathbf{F}^\top \end{bmatrix}$$

$$= \begin{bmatrix} \mathbf{R} & \mathbf{0} \\ \mathbf{0} & \mathbf{F} \end{bmatrix}\begin{bmatrix} (\mathbf{R}^\top\mathbf{R})^{-1} & \mathbf{0} \\ \mathbf{0} & (\mathbf{F}^\top\mathbf{F})^{-1} \end{bmatrix}\begin{bmatrix} \mathbf{R}^\top & \mathbf{0} \\ \mathbf{0} & \mathbf{F}^\top \end{bmatrix}.$$

The remaining Projection matrices can be obtained by similar computations.

Additionally, we have $\mathbf{X}(\mathbf{I} - \mathbf{P}) = (\mathbf{I} - \mathbf{P})\mathbf{X} = \mathbf{X}(\mathbf{I} - \mathbf{X}(\mathbf{X}^\top\mathbf{X})^{-1}\mathbf{X}^\top) = 0$.

Moreover, since $\mathbf{A}$ is a submatrix of $\mathbf{X}$, it can be represented as $\mathbf{A} = \mathbf{XC}$ for some selective matrix $\mathbf{C}$. Therefore, we have:

$$\mathbf{PA} = \mathbf{X}\left(\mathbf{X}^\top\mathbf{X}\right)^{-1}\mathbf{X}^\top\mathbf{XC} = \mathbf{XC} = \mathbf{A}.$$

Meanwhile, it also holds that

$$\mathbf{PP}_A = \mathbf{X}(\mathbf{X}^\top\mathbf{X})^{-1}\mathbf{X}^\top\mathbf{XC}(\mathbf{C}^\top\mathbf{X}^\top\mathbf{XC})^{-1}\mathbf{C}^\top\mathbf{X}^\top = \mathbf{XC}(\mathbf{C}^\top\mathbf{X}^\top\mathbf{XC})^{-1}\mathbf{C}^\top\mathbf{X}^{\cdot\top} = \mathbf{P}_A.$$

$\mathbf{X}_r$ and $\mathbf{X}_f$ are submatrices of $\mathbf{X}$, each with disjoint spaces. The projection of $\mathbf{X}_r$ onto the space of $\mathbf{X}_f$ should be zero.

$$\mathbf{X}_r\mathbf{P}_f = \mathbf{X}_r\mathbf{X}_f(\mathbf{X}_f^\top\mathbf{X}_f)^{-1}\mathbf{X}_f^\top = 0.$$

$\square$

**Corollary B.2** (Minimum Norm Solution 1). *Let $\mathbf{P}, \mathbf{P}_r, \mathbf{P}_f, \mathbf{P}_t$ be the corresponding projection operator for $\mathbf{X}, \mathbf{X}_r, \mathbf{X}_f, \mathbf{X}_t$ respectively. Then, the solution to the optimization problem Equation (1), Equation (2) and Equation (3) can be represented by:*

1. *Under Assumption 3.1, $\mathbf{w}_o = \mathbf{Pw}_*$, $\mathbf{w}_t = (\mathbf{I} - \mathbf{P}_t)\mathbf{w}_o + \mathbf{P}_t\mathbf{w}_*^r$, and $\mathbf{w}_g = \mathbf{P}_r\mathbf{w}_*^r$;*

2. *Under Assumption 3.3, $\mathbf{w}_o = \mathbf{Pw}_*$, $\mathbf{w}_t = (\mathbf{I} - \mathbf{P}_t)\mathbf{w}_o + \mathbf{P}_t(\mathbf{w}_*^r + \mathbf{w}_*^{lap})$, and $\mathbf{w}_g = \mathbf{P}_r(\mathbf{w}_*^r + \mathbf{w}_*^{lap})$;*

3. *$\mathbf{X}_r^\top\mathbf{w}_*^f = 0$ and $\mathbf{X}_f^\top\mathbf{w}_*^r = 0$.*

**Proof of Corollary B.2.** According to the method of Lagrange multipliers and the problem setup, it is easy to obtain the first two conclusions. For the last one, we have:

$$\mathbf{X}_r^\top\mathbf{w}_*^f = [\mathbf{R}^\top, \mathbf{0}]\mathbf{w}_*^f = 0 \quad \text{and} \quad \mathbf{X}_f^\top\mathbf{w}_*^r = [\mathbf{0}, \mathbf{F}^\top]\mathbf{w}_*^r = 0.$$

$\square$

### B.2 PROOF OF THEOREM 3.2

Let us first focus on the performance of the golden model. Based on the definition of unlearning accuracy and retaining accuracy, we have

$$\text{RL:} \quad L(\mathbf{w}_g, D_r) = \frac{1}{n_r}\|\mathbf{X}_r^\top\mathbf{w}_g - \mathbf{y}_r\|^2 = \frac{1}{n_r}\|\mathbf{X}_r^\top\mathbf{P}_r\mathbf{w}_*^r - \mathbf{X}_r^\top\mathbf{w}_*^r\|^2 = \frac{1}{n_r}\|\mathbf{X}_r^\top(\mathbf{P}_r - \mathbf{I})\mathbf{w}_*^r\|^2 = 0,$$

where the second equality arises from the model setting and Proposition B.2, while the penultimate equality is due to the properties of the projection matrix. According to Corollary B.1, we have

$$\text{UL:} \quad L(\mathbf{w}_g, D_f) = \frac{1}{n_f}\|\mathbf{X}_f^\top\mathbf{w}_g - \mathbf{y}_f\|^2 = \frac{1}{n_f}\|\mathbf{X}_f^\top\mathbf{P}_r\mathbf{w}_*^r - \mathbf{X}_f^\top\mathbf{w}_*^f\|^2$$

$$= \frac{1}{n_f}\left\|[\mathbf{0}, \mathbf{F}^\top]\begin{bmatrix} \mathbf{R}(\mathbf{R}^\top\mathbf{R})^{-1}\mathbf{R}^\top & \mathbf{0} \\ \mathbf{0} & \mathbf{0} \end{bmatrix}\mathbf{w}_*^r - \mathbf{X}_f^\top\mathbf{w}_*^f\right\|^2 = \frac{1}{n_f}\|\mathbf{X}_f^\top\mathbf{w}_*^f\|^2.$$

Similarly, for the fine-tuning model, it holds that

$$\text{RL:} \quad L(\mathbf{w}_t, D_r) = \frac{1}{n_r}\|\mathbf{X}_r^\top \mathbf{w}_t - \mathbf{y}_r\|^2 = \frac{1}{n_r}\|\mathbf{X}_r^\top((\mathbf{I} - \mathbf{P}_t)\mathbf{w}_o + \mathbf{P}_t\mathbf{w}_*^r) - \mathbf{X}_r^\top \mathbf{w}_*^r\|^2$$

$$= \frac{1}{n_r}\|\mathbf{X}_r^\top((\mathbf{I} - \mathbf{P}_t)\mathbf{P}(\mathbf{w}_*^r + \mathbf{w}_*^f) + \mathbf{P}_t\mathbf{w}_*^r) - \mathbf{X}_r^\top \mathbf{w}_*^r\|^2$$

$$= \frac{1}{n_r}\|\mathbf{X}_r^\top(\mathbf{P}\mathbf{w}_*^r + (\mathbf{P} - \mathbf{P}_t)\mathbf{w}_*^f) - \mathbf{X}_r^\top \mathbf{w}_*^r\|^2$$

$$= 0.$$

$$\text{UL:} \quad L(\mathbf{w}_t, D_f) = \frac{1}{n_f}\|\mathbf{X}_f^\top \mathbf{w}_t - \mathbf{y}_f\|^2 = \frac{1}{n_f}\|\mathbf{X}_f^\top((\mathbf{I} - \mathbf{P}_t)\mathbf{w}_o + \mathbf{P}_t\mathbf{w}_*^r) - \mathbf{X}_f^\top \mathbf{w}_*^f\|^2$$

$$= \frac{1}{n_f}\|\mathbf{X}_f^\top[(\mathbf{I} - \mathbf{P}_t)\mathbf{P}\mathbf{w}_* + \mathbf{P}_t\mathbf{w}_*^r - \mathbf{w}_*^f]\|^2$$

$$= \frac{1}{n_f}\|\mathbf{X}_f^\top[(\mathbf{I} - \mathbf{P}_t)\mathbf{P} + \mathbf{P}_t]\mathbf{w}_*^r + \mathbf{X}_f^\top[(\mathbf{I} - \mathbf{P}_t)\mathbf{P} - \mathbf{I}]\mathbf{w}_*^f]\|^2$$

$$= \frac{1}{n_f}\|\mathbf{X}_f^\top \mathbf{P}\mathbf{w}_*^r + \mathbf{X}_f^\top \mathbf{P}\mathbf{w}_*^f - \mathbf{X}_f^\top \mathbf{w}_*^f]\|^2$$

$$= 0,$$

where the penultimate equality comes from $\mathbf{X}_f^\top \mathbf{P}_t = \mathbf{X}_f^\top \mathbf{P}_r = 0$, and the last equality follows from $\mathbf{X}_f^\top \mathbf{P} = \mathbf{X}_f^\top$.

### B.3 PROOF OF THEOREM 3.4

Due to the assumption of overlapping features, the projection properties of the dataset matrix will be slightly different. Specifically, it holds that:

**Corollary B.3** (Projection Matrix properties′)**.** *Let* $\mathbf{P} = \mathbf{X}(\mathbf{X}^\top \mathbf{X})^{-1}\mathbf{X}^\top, \mathbf{P}_r, \mathbf{P}_f, \mathbf{P}_t$ *be the corresponding projection operator for* $\mathbf{X}, \mathbf{X}_r, \mathbf{X}_f, \mathbf{X}_t$ *respectively. Under Assumption 3.3, it holds that:*

2. $\mathbf{P}_r = \begin{bmatrix} \mathbf{R}(\mathbf{R}^\top\mathbf{R} + \mathbf{L}_1^\top\mathbf{L}_1)^{-1}\mathbf{R}^\top & \mathbf{R}(\mathbf{R}^\top\mathbf{R} + \mathbf{L}_1^\top\mathbf{L}_1)^{-1}\mathbf{L}_1^\top & \mathbf{0} \\ \mathbf{L}_1(\mathbf{R}^\top\mathbf{R} + \mathbf{L}_1^\top\mathbf{L}_1)^{-1}\mathbf{R}^\top & \mathbf{L}_1(\mathbf{R}^\top\mathbf{R} + \mathbf{L}_1^\top\mathbf{L}_1)^{-1}\mathbf{L}_1^\top & \mathbf{0} \\ \mathbf{0} & \mathbf{0} & \mathbf{0} \end{bmatrix};$

3. $\mathbf{P}_f = \begin{bmatrix} \mathbf{0} & \mathbf{0} & \mathbf{0} \\ \mathbf{0} & \mathbf{L}_2(\mathbf{F}^\top\mathbf{F} + \mathbf{L}_2^\top\mathbf{L}_2)^{-1}\mathbf{L}_2^\top & \mathbf{L}_2(\mathbf{F}^\top\mathbf{F} + \mathbf{L}_2^\top\mathbf{L}_2)^{-1}\mathbf{F}^\top \\ \mathbf{0} & \mathbf{F}(\mathbf{F}^\top\mathbf{F} + \mathbf{L}_2^\top\mathbf{L}_2)^{-1}\mathbf{L}_2^\top & \mathbf{F}(\mathbf{F}^\top\mathbf{F} + \mathbf{L}_2^\top\mathbf{L}_2)^{-1}\mathbf{F}^\top \end{bmatrix};$

3. $\mathbf{X}(\mathbf{I} - \mathbf{P}) = (\mathbf{I} - \mathbf{P})\mathbf{X} = 0$, *and the conclusion also holds for* $\mathbf{P}_r, \mathbf{P}_f, \mathbf{P}_t$ *with* $\mathbf{X}_r, \mathbf{X}_f, \mathbf{X}_t$ *respectively;*

4. *For any matrix* $\mathbf{A}$ *is the submatrix of* $\mathbf{X}$, *it holds that* $\mathbf{A} = \mathbf{P}\mathbf{A}$, *where* $\mathbf{P}$ *is the projection space of* $\mathbf{X}$. *Moreover, if* $\mathbf{P}_A$ *is the projection space of* $\mathbf{A}$, *it holds that* $\mathbf{P}\mathbf{P}_A = \mathbf{P}_A$.

**Proof of Corollary B.3.** Proof of Corollary B.3 follows the proof of Corollary B.1 directly. □

Now we are ready to go through the proof of Theorem 3.4. Similar to the non-overlapping case, the golden model holds that

$$\text{RL:} \quad L(\mathbf{w}_g, D_r) = \frac{1}{n_r}\|\mathbf{X}_r^\top \mathbf{w}_g - \mathbf{y}_r\|^2 = \frac{1}{n_r}\|\mathbf{X}_r^\top \mathbf{P}_r(\mathbf{w}_*^r + \mathbf{w}_*^{lap}) - \mathbf{X}_r^\top(\mathbf{w}_*^r + \mathbf{w}_*^{lap})\|^2$$

$$= \frac{1}{n_r}\|\mathbf{X}_r^\top(\mathbf{P}_r - \mathbf{I})(\mathbf{w}_*^r + \mathbf{w}_*^{lap})\|^2 = 0,$$

where the second equality also arises from the model setting and Proposition B.2, while the penultimate equality is due to the properties of the projection matrix. According to Corollary B.3, we have

$$\text{UL:} \quad L(\mathbf{w}_g, D_f) = \frac{1}{n_f}\|\mathbf{X}_f^\top \mathbf{w}_g - \mathbf{y}_f\|^2 = \frac{1}{n_f}\|\mathbf{X}_f^\top \mathbf{P}_r(\mathbf{w}_*^r + \mathbf{w}_*^{lap}) - \mathbf{X}_f^\top(\mathbf{w}_*^f + \mathbf{w}_*^{lap})\|^2$$

where $\mathbf{X}_f^\top \mathbf{P}_r \mathbf{w}_*^r$ and $\mathbf{X}_f^\top \mathbf{P}_r \mathbf{w}_*^{lap}$ follows that

$$\mathbf{X}_f^\top \mathbf{P}_r \mathbf{w}_*^r = [\mathbf{0}, \mathbf{L}_2^\top, \mathbf{F}^\top] \begin{bmatrix} \mathbf{R}(\mathbf{R}^\top\mathbf{R} + \mathbf{L}_1^\top\mathbf{L}_1)^{-1}\mathbf{R}^\top & \mathbf{R}(\mathbf{R}^\top\mathbf{R} + \mathbf{L}_1^\top\mathbf{L}_1)^{-1}\mathbf{L}_1^\top & \mathbf{0} \\ \mathbf{L}_1(\mathbf{R}^\top\mathbf{R} + \mathbf{L}_1^\top\mathbf{L}_1)^{-1}\mathbf{R}^\top & \mathbf{L}_1(\mathbf{R}^\top\mathbf{R} + \mathbf{L}_1^\top\mathbf{L}_1)^{-1}\mathbf{L}_1^\top & \mathbf{0} \\ \mathbf{0} & \mathbf{0} & \mathbf{0} \end{bmatrix} \begin{bmatrix} \square \\ \mathbf{0} \\ \mathbf{0} \end{bmatrix}$$

$$= \mathbf{L}_2^\top \mathbf{L}_1 (\mathbf{R}^\top\mathbf{R} + \mathbf{L}_1^\top\mathbf{L}_1)^{-1}\mathbf{R}^\top \mathbf{w}_*^r$$

and

$$\mathbf{X}_f^\top \mathbf{P}_r \mathbf{w}_*^{lap} = [\mathbf{0}, \mathbf{L}_2^\top, \mathbf{F}^\top] \begin{bmatrix} \mathbf{R}(\mathbf{R}^\top\mathbf{R} + \mathbf{L}_1^\top\mathbf{L}_1)^{-1}\mathbf{R}^\top & \mathbf{R}(\mathbf{R}^\top\mathbf{R} + \mathbf{L}_1^\top\mathbf{L}_1)^{-1}\mathbf{L}_1^\top & \mathbf{0} \\ \mathbf{L}_1(\mathbf{R}^\top\mathbf{R} + \mathbf{L}_1^\top\mathbf{L}_1)^{-1}\mathbf{R}^\top & \mathbf{L}_1(\mathbf{R}^\top\mathbf{R} + \mathbf{L}_1^\top\mathbf{L}_1)^{-1}\mathbf{L}_1^\top & \mathbf{0} \\ \mathbf{0} & \mathbf{0} & \mathbf{0} \end{bmatrix} \begin{bmatrix} \mathbf{0} \\ \square \\ \mathbf{0} \end{bmatrix}$$

$$= \mathbf{L}_2^\top \mathbf{L}_1 (\mathbf{R}^\top\mathbf{R} + \mathbf{L}_1^\top\mathbf{L}_1)^{-1}\mathbf{L}_1^\top \mathbf{w}_*^{lap}.$$

For the fine-tuning, the retaining accuracy follows:

$$\begin{aligned}
\text{RL:} \quad L(\mathbf{w}_t, D_r) &= \frac{1}{n_r}\|\mathbf{X}_r^\top \mathbf{w}_t - \mathbf{y}_r\|^2 \\
&= \frac{1}{n_r}\|\mathbf{X}_r^\top((\mathbf{I} - \mathbf{P}_t)\mathbf{w}_o + \mathbf{P}_t(\mathbf{w}_*^r + \mathbf{w}_*^{lap})) - \mathbf{X}_r^\top(\mathbf{w}_*^r + \mathbf{w}_*^{lap})\|^2 \\
&= \frac{1}{n_r}\|\mathbf{X}_r^\top(\mathbf{I} - \mathbf{P}_t)(\mathbf{w}_o - \mathbf{w}_*^r - \mathbf{w}_*^{lap})\|^2 \\
&= \frac{1}{n_r}\|\mathbf{X}_r^\top(\mathbf{I} - \mathbf{P}_t)(\mathbf{P} - \mathbf{I})(\mathbf{w}_*^r + \mathbf{w}_*^{lap})\|^2 \\
&= 0.
\end{aligned}$$

The last equality derives from that the facts the projection matrix is commutative matrix and the last property holds in Corollary B.3. For the unlearning accuracy, it holds that

$$\begin{aligned}
\text{UL:} \quad L(\mathbf{w}_t, D_f) &= \frac{1}{n_f}\|\mathbf{X}_f^\top \mathbf{w}_t - \mathbf{y}_f\|^2 \\
&= \frac{1}{n_f}\|\mathbf{X}_f^\top((\mathbf{I} - \mathbf{P}_t)\mathbf{w}_o + \mathbf{P}_t(\mathbf{w}_*^r + \mathbf{w}_*^{lap})) - \mathbf{X}_f^\top(\mathbf{w}_*^f + \mathbf{w}_*^{lap})\|^2 \\
&= \frac{1}{n_f}\|\mathbf{X}_f^\top((\mathbf{I} - \mathbf{P}_t)(\mathbf{P}(\mathbf{w}_*^r + \mathbf{w}_*^{lap} + \mathbf{w}_*^f)) + \mathbf{P}_t(\mathbf{w}_*^r + \mathbf{w}_*^{lap})) - \mathbf{X}_f^\top(\mathbf{w}_*^f + \mathbf{w}_*^{lap})\|^2 \\
&= \frac{1}{n_f}\|\mathbf{X}_f^\top((\mathbf{P} - \mathbf{P}_t)(\mathbf{w}_*^r + \mathbf{w}_*^{lap} + \mathbf{w}_*^f)) + \mathbf{P}_t(\mathbf{w}_*^r + \mathbf{w}_*^{lap})) - \mathbf{X}_f^\top(\mathbf{w}_*^f + \mathbf{w}_*^{lap})\|^2 \\
&= \frac{1}{n_f}\|\mathbf{X}_f^\top[(\mathbf{P} - \mathbf{I})\mathbf{w}_*^{lap} + \mathbf{P}\mathbf{w}_*^r + (\mathbf{P} - \mathbf{I} - \mathbf{P}_t)\mathbf{w}_*^f]\|^2 \\
&= \frac{1}{n_f}\|\mathbf{X}_f^\top \mathbf{P}_t \mathbf{w}_*^f]\|^2 \\
&= 0,
\end{aligned}$$

where the penultimate equality is due to Corollary B.3 and the last equality comes from the fact $\mathbf{X}_t$ enjoys the same data structure as $\mathbf{X}_t$ such that:

$$\mathbf{X}_f^\top \mathbf{P}_t \mathbf{w}_*^f$$

$$= [\mathbf{0}, \mathbf{L}_2^\top, \mathbf{F}^\top] \begin{bmatrix} \mathbf{R}_T(\mathbf{R}_T^\top \mathbf{R}_T + \mathbf{L}_{1T}^\top \mathbf{L}_{1T})^{-1}\mathbf{R}_T^\top & \mathbf{R}_T(\mathbf{R}_T^\top \mathbf{R}_T + \mathbf{L}_{1T}^\top \mathbf{L}_{1T})^{-1}\mathbf{L}_{1T}^\top & \mathbf{0} \\ \mathbf{L}_{1T}(\mathbf{R}_T^\top \mathbf{R}_T + \mathbf{L}_{1T}^\top \mathbf{L}_{1T})^{-1}\mathbf{R}_T^\top & \mathbf{L}_{1T}(\mathbf{R}_T^\top \mathbf{R}_T + \mathbf{L}_{1T}^\top \mathbf{L}_{1T})^{-1}\mathbf{L}_{1T}^\top & \mathbf{0} \\ \mathbf{0} & \mathbf{0} & \mathbf{0} \end{bmatrix} \begin{bmatrix} \mathbf{0} \\ \mathbf{0} \\ \square \end{bmatrix}$$

$$= [\mathbf{L}_2^\top \mathbf{L}_{1T}(\mathbf{R}_T^\top \mathbf{R}_T + \mathbf{L}_{1T}^\top \mathbf{L}_{1T})^{-1}\mathbf{R}_T^\top, \mathbf{L}_2^\top \mathbf{L}_{1T}(\mathbf{R}_T^\top \mathbf{R}_T + \mathbf{L}_{1T}^\top \mathbf{L}_{1T})^{-1}\mathbf{L}_{1T}^\top, 0] \begin{bmatrix} \mathbf{0} \\ \mathbf{0} \\ \square \end{bmatrix}$$

$$= 0.$$

### B.4 PROOF OF THEOREM 4.1

For the non-overlapping case, we have that the retaining accuracy follows:

$$\text{RL:} \quad L(\mathbf{w}_t, D_r) = \frac{1}{n_r}\|\mathbf{X}_r^\top \mathbf{w}_t - \mathbf{y}_r\|^2 = \frac{1}{n_r}\|\mathbf{X}_r^\top((\mathbf{I} - \mathbf{P}_t)\hat{\mathbf{w}}_o + \mathbf{P}_t \mathbf{w}_*^r) - \mathbf{X}_r^\top \mathbf{w}_*^r\|^2$$

$$= \frac{1}{n_r}\|\mathbf{X}_r^\top(\mathbf{I} - \mathbf{P}_t)(\hat{\mathbf{w}}_o - \mathbf{w}_*^r)\|^2$$

$$= \frac{1}{n_r}\|\mathbf{X}_r^\top(\mathbf{I} - \mathbf{P}_t)(\mathbf{P} - \mathbf{I})\mathbf{w}_*^r\|^2 = 0.$$

For the unlearning accuracy, it holds that

$$\text{UL:} \quad L(\mathbf{w}_t, D_f) = \frac{1}{n_f}\|\mathbf{X}_f^\top \mathbf{w}_t - \mathbf{y}_f\|^2 = \frac{1}{n_f}\|\mathbf{X}_f^\top((\mathbf{I} - \mathbf{P}_t)\hat{\mathbf{w}}_o + \mathbf{P}_t \mathbf{w}_*^r) - \mathbf{X}_f^\top \mathbf{w}_*^f\|^2$$

$$= \frac{1}{n_f}\|\mathbf{X}_f^\top[(\mathbf{I} - \mathbf{P}_t)\mathbf{P}\mathbf{w}_*^r + \mathbf{P}_t \mathbf{w}_*^r - \mathbf{w}_*^f]\|^2$$

$$= \frac{1}{n_f}\|\mathbf{X}_f^\top[(\mathbf{I} - \mathbf{P}_t)\mathbf{P} + \mathbf{P}_t]\mathbf{w}_*^r - \mathbf{X}_f^\top \mathbf{w}_*^f]\|^2$$

$$= \frac{1}{n_f}\|\mathbf{X}_f^\top \mathbf{P}\mathbf{w}_*^r - \mathbf{X}_f^\top \mathbf{w}_*^f]\|^2$$

$$= \frac{1}{n_f}\|\mathbf{w}_*^f]\|_{\mathbf{X}_f \mathbf{X}_f^\top}^2,$$

For the overlapping case, it holds that

$$\text{RL:} \quad L(\mathbf{w}_t, D_r) = \frac{1}{n_r}\|\mathbf{X}_r^\top \mathbf{w}_t - \mathbf{y}_r\|^2$$

$$= \frac{1}{n_r}\|\mathbf{X}_r^\top((\mathbf{I} - \mathbf{P}_t)\hat{\mathbf{w}}_o + \mathbf{P}_t(\mathbf{w}_*^r + \mathbf{w}_*^{lap})) - \mathbf{X}_r^\top(\mathbf{w}_*^r + \mathbf{w}_*^{lap})\|^2$$

$$= \frac{1}{n_r}\|\mathbf{X}_r^\top(\mathbf{I} - \mathbf{P}_t)(\hat{\mathbf{w}}_o - \mathbf{w}_*^r - \mathbf{w}_*^{lap})\|^2$$

$$= \frac{1}{n_r}\|\mathbf{X}_r^\top(\mathbf{I} - \mathbf{P}_t)(\mathbf{P} - \mathbf{I})(\mathbf{w}_*^r - \mathbf{w}_*^{lap})\|^2 = 0.$$

$$\text{UL:} \quad L(\mathbf{w}_t, D_f) = \frac{1}{n_f}\|\mathbf{X}_f^\top \mathbf{w}_t - \mathbf{y}_f\|^2$$

$$= \frac{1}{n_f}\|\mathbf{X}_f^\top((\mathbf{I} - \mathbf{P}_t)\hat{\mathbf{w}}_o + \mathbf{P}_t(\mathbf{w}_*^r + \mathbf{w}_*^{lap})) - \mathbf{X}_f^\top(\mathbf{w}_*^f + \mathbf{w}_*^{lap})\|^2$$

$$= \frac{1}{n_f}\|\mathbf{X}_f^\top((\mathbf{I} - \mathbf{P}_t)(\mathbf{P}(\mathbf{w}_*^r + \mathbf{w}_*^{lap})) + \mathbf{P}_t(\mathbf{w}_*^r + \mathbf{w}_*^{lap})) - \mathbf{X}_f^\top(\mathbf{w}_*^f + \mathbf{w}_*^{lap})\|^2$$

$$= \frac{1}{n_f}\|\mathbf{X}_f^\top((\mathbf{P} - \mathbf{P}_t)(\mathbf{w}_*^r + \mathbf{w}_*^{lap})) + \mathbf{P}_t(\mathbf{w}_*^r + \mathbf{w}_*^{lap})) - \mathbf{X}_f^\top(\mathbf{w}_*^f + \mathbf{w}_*^{lap})\|^2$$

$$= \frac{1}{n_f}\|\mathbf{X}_f^\top[(\mathbf{P} - \mathbf{I})\mathbf{w}_*^{lap} + \mathbf{P}\mathbf{w}_*^r - \mathbf{w}_*^f]\|^2$$

$$= \frac{1}{n_f}\|\mathbf{P}(\mathbf{w}_*^r + \mathbf{w}_*^{lap}) - (\mathbf{w}_*^f + \mathbf{w}_*^{lap})\|_{\mathbf{X}_f \mathbf{X}_f^\top}^2.$$

The proof is then complete.

