# OpenReview forum: "Why Fine-Tuning Struggles with Forgetting in Machine Unlearning? Theoretical Insights and a Remedial Approach"
_ICLR.cc/2025/Conference — Submitted to ICLR 2025_

### Official Review · Reviewer_J9SB · 2024-11-02

**Soundness:** 3
**Presentation:** 4
**Contribution:** 3
**Rating:** 6
**Confidence:** 5

**Summary:**

This paper investigates machine unlearning, which aims to protect user privacy by removing specific data from trained models in response to data deletion requests. The authors examine why fine-tuning often fails to fully erase targeted data. They consider over-parameterized linear regression in the case of overlapping and no overlapping features. They propose a regularization term that diminishes the role of forgetting. Experimental results on both synthetic and real-world datasets validate that this regularization approach significantly enhances unlearning performance.

**Strengths:**

Studying unlearning from the perspective of overparametrizied regression is a great concept. This setting (even though questionable in practice) allows to perform theoretical analyses.

The entire concept of introducing a regularization term to unlearning is very sound and novel.

The experimental results show improvements if such a term is included.

**Weaknesses:**

The setting for the analyses is simplistic. It would be great to consider more general cases (for example strongly convex).

I think the vast majority of the practical cases consider 100% overlapping cases which puts in question the bulk of the analyses.

The distinct features section is a special case of the overlapping section and thus it should be omitted. I think the distinct features results are not stronger and thus they are a 'strict' special case.

Option B is 'void' if all of the features are overlapping which captures the majority of the use cases.

**Questions:**

1. Why bothering with the case of non overlapping features? While such cases sometimes occur in FL they are a much more seldom occurrence in standard ML.
2 Can the analyses be done for the strongly convex case? It seems it would require a completely different approach since a closed form expression is not available in such a case. What about a 2 layer linear network?

---

> ### Author Response · Authors · 2024-11-20
> **Reponse**
>
> We sincerely thank the Reviewer J9SB for the valuable time and positive comments on our work! We hope our following answers could address your concerns.
>
> **Response to Weakness 1:**
> Thanks for the comment! Overparameterized linear models are widely adopted as a foundational framework for studying learning problems (e.g., transfer learning [1], continual learning [2], In-context learning [3,4]) and can be extended to more general settings such as neural tangent kernel (NTK) analysis. While the setting of overparameterized linear regression is indeed simplistic, it provides a valuable starting point for analyzing training dynamics, capturing the trajectory of learning rather than just upper or lower bounds.
> In this work, we start with the overparameterized linear regression model and hope to extend the analysis to more general cases such as multi-layer neural network in the future.
>
> **Response to Weakness 2:**
> Thank you for your comment! Our primary motivation stems from the empirical observation that while fine-tuning can maintain model utility on the remaining data, it often struggles to effectively forget targeted data. To provide theoretical insights into this phenomenon, we begin with a simpler case where the dataset is completely separable.
>
> For example, consider a dataset containing two categories: bananas and cars. Bananas have a distinct feature like "elongated shape," and cars have a unique feature like "mirrors." These features are entirely distinct. We then extend this setup to include overlapping features, such as color, where both bananas and cars might share a feature like "yellow."
>
> We recognize that most practical cases involve overlapping features. However, presenting only the overlapping case would relegate the distinct case to a special case, disrupting the presentation flow. Our goal is to demonstrate that even in the extreme case of distinct features, fine-tuning still fails to unlearn. Therefore, starting with a simpler, distinct feature setup allows us to build a foundation to explain why fine-tuning fails to unlearn. From this base, we advance the analysis to handle more complex, overlapping cases.
>
> **Response to Weakness 3:**
> Yes, the scenario you mention aligns with our findings in Theorem 4.1, which demonstrates that discarding overlapping features leads to an increase in the retaining loss. To address this, in the next section, we propose using both retain loss and unlearn loss as objective functions, allowing the model to effectively balance retaining overlapping features while unlearning targeted information.
>
> **Response to Question 1:**
> As mentioned in our previous response, we discuss the case of nonoverlapping features as a starting point to build intuition about the phenomenon of fine-tuning failing to unlearn. While this case may be less common in standard ML, it provides a simplified setup that allows us to derive theoretical insights and establish a foundation for analyzing more complex overlapping scenarios.
>
> Our current work focuses on overparameterized linear regression as a framework for machine unlearning. Extending this analysis to strongly convex settings or a two-layer linear network would indeed require different techniques, as closed-form solutions are not readily available. We agree that such extensions are valuable and will consider them as part of future work. Thank you for the suggestion!
>
> References:
>
> [1] Wu, Jingfeng, et al. "The power and limitation of pretraining-finetuning for linear regression under covariate shift." Advances in Neural Information Processing Systems 35 (2022): 33041-33053.
>
> [2] Ding, Meng, et al. "Understanding Forgetting in Continual Learning with Linear Regression." arXiv preprint arXiv:2405.17583 (2024).
>
> [3] Wu, Jingfeng, et al. "How Many Pretraining Tasks Are Needed for In-Context Learning of Linear Regression?." arXiv preprint arXiv:2310.08391 (2023).
>
> [4] Chen, Xingwu, Lei Zhao, and Difan Zou. "How transformers utilize multi-head attention in in-context learning? a case study on sparse linear regression." arXiv preprint arXiv:2408.04532 (2024).

---

> > ### Author Response · Authors · 2024-11-24
> >
> > Thank you once again for your initial review! We have carefully addressed your comments and provided a detailed response. As the discussion period deadline approaches, we would greatly value your feedback on our response.

---

### Official Review · Reviewer_NAAU · 2024-11-02

**Soundness:** 1
**Presentation:** 3
**Contribution:** 1
**Rating:** 3
**Confidence:** 3

**Summary:**

This paper works on why the current fine-tuning unlearning method cannot perform well in many unlearning tasks. This paper provides a theoretical analysis within a linear regression framework to show when fine-tuning retains model performance on remaining data, it cannot fully remove the influence of the forgetting data. Then this paper proposes a discriminative regularization term to close the performance gap between fine-tuned model and retrained model. The experimental results validate the effectiveness of this approach in improving unlearning accuracy.

**Strengths:**

1. The topic of this paper is quite interesting. The fine-tuning approach is one of the mainstream approaches to unlearning. However, such methods are usually unstable across different unlearning tasks and datasets. Thus, the research on why it can fail is meaninful.
2. This paper provides a theoretical analysis of the linear regression model for the analysis.
3. The experimental results can clearly show the performance improvements compared with the other fine-tuning methods.

**Weaknesses:**

The total contribution of the paper is not enough:

1. Theoretical Analysis: Target on theorem 3.2 and 3.4: this paper claims that the MSE on remaining data and forgetting data keep 0 during fine-tuning for overparameterized models. However, in real-world datasets, the model cannot fit the training data perfectly, and the two theorems are hard to extend to other larger models and datasets. In addition, such analysis is based on a regression model, while the following experiment part is mainly based on classification tasks. Whether the theoretical analysis on regression can be extended to classification still needs to be proved.

2. Discriminative Regularization: This paper does not explicitly show the loss of Inverse CE-FT. Is it simply to remove the hyperparameter $\alpha$ from the second term to the first term? If so, I cannot find a significant difference between Inverse CE-FT and original CE-FT. In addition, regarding the loss function of KL-FT, many other methods have tried to incorporate KL loss to align the output logits [1] or interlayer embeddings [2, 3]. Therefore, the proposed Discriminative Regularization does not show any improvement compared with previous works.

3. Experiment results: This paper only conducts experiments on single-class unlearning (classes 3,6 and 9). In addition, this paper only compares the proposed method with naive fine-tuning and the loss proposed in [4]. It is not sufficient to prove the effectiveness of the proposed methods. This paper can include more SOTA unlearning methods in the recent two years and compare unlearning results in more complex settings like random sample unlearning or backdoor attack unlearning.

4. The technical part of this paper needs to be improved. Some notations need to be further checked, for example, $1-n_f$ in line 194.

[1]  Chundawat, Vikram S., et al. "Can bad teaching induce forgetting? unlearning in deep networks using an incompetent teacher." Proceedings of the AAAI Conference on Artificial Intelligence. Vol. 37. No. 6. 2023.

[2]  Chundawat, Vikram S., et al. "Zero-shot machine unlearning." IEEE Transactions on Information Forensics and Security 18 (2023): 2345-2354.

[3]  Shen, Shaofei, et al. "CaMU: Disentangling Causal Effects in Deep Model Unlearning." Proceedings of the 2024 SIAM International Conference on Data Mining (SDM). Society for Industrial and Applied Mathematics, 2024.

[4] Fan, Chongyu, et al. "Salun: Empowering machine unlearning via gradient-based weight saliency in both image classification and generation." arXiv preprint arXiv:2310.12508 (2023).

**Questions:**

1.  What do the distinct and overlap features mean? Could the author give some examples to explain it?

2.  Considering that this paper mainly conducts class-wise unlearning experiments. How do different methods perform under the evaluation of relearn time [1]?

[1]  Chundawat, Vikram S., et al. "Zero-shot machine unlearning." IEEE Transactions on Information Forensics and Security 18 (2023): 2345-2354.

---

> ### Author Response · Authors · 2024-11-20
> **Response**
>
> We thank the Reviewer NAAU for the detailed review. We hope our following answers can address your concerns.
>
> **Response to Weakness 1:** Thanks for the point! Overparameterized linear models are widely adopted as a foundational framework for studying learning problems (e.g., transfer learning [1], continual learning [2], In-context learning [3,4]) and can be extended to more general settings such as neural tangent kernel (NTK) analysis. While the setting of overparameterized linear regression is indeed simplistic, it provides a valuable starting point for analyzing training dynamics, capturing the trajectory of learning rather than just upper or lower bounds.
> As for the focus on regression versus classification, although we use linear regression to analyze the fine-tuning process, our
> assumptions about the data structure are based on a classification problem, which is why our experiments focus on class-wise forgetting. In this work, we start with the overparameterized linear regression model and hope to extend the analysis to more general cases such as multi-layer neural network in the future.
>
> **Response to Weakness 2:** Thank you for pointing out this!  We acknowledge that there is limited distinction between ICE-FT approach and CE-FT, primarily differing in the choice of the parameter $\alpha$. However, based on the analysis in Sections 3 and 4, we aim to understand the empirical phenomenon of why naive fine-tuning fails to forget. Therefore, Theorem 4.1 provides the rationale for selecting the parameter $\alpha \in(0,1]$ for the unlearning loss (the design of ICE-FT), whereas in [5], the constraint $\alpha \in(0,1]$ is applied to the retain loss. **While our objective formulation may appear similar to existing work, our novelty/contribution goes beyond the design of the regularizer.** In response to this weakness, we have rewritten Section 5 to modify our previous statement. Please check it. Thank you for the comment!
>
> **Response to Weakness 3:**
> We appreciate the Reviewer’s comment and agree that our experiments do not cover more complex unlearning settings or comparisons with a wide range of SOTA methods. However, the primary focus of our paper is to understand the empirical phenomenon of why naive fine-tuning fails to forget, as analyzed in Sections 3 and 4. The experiments are designed to validate the conclusions derived from our theoretical framework rather than to benchmark against numerous unlearning methods.
>
> **Response to Weakness 4:**
> Thank you for pointing this out! We have corrected this notation in our revised version.
>
> **Response to Question 1:**
> Distinct features refer to features unique to either forgetting data or retaining data, while overlapping features are shared between the two. For example, consider a dataset containing two categories: bananas and cars. Bananas have a distinct feature like an "elongated shape," and cars have a unique feature like "mirrors." These features are entirely distinct. We then extend this setup to include overlapping features, such as color, where both bananas and cars might share a feature like "yellow."
>
> **Response to Question 2:**
> Thank you for your question! In the machine unlearning community, various evaluation metrics and settings exist, including relearn time, as mentioned by the Reviewer. Relearn time, as defined in [6], measures the time needed to achieve a margin of $\alpha \\%$ around the original accuracy in a zero-shot unlearning setting, where $\alpha \%$ reflects the accuracy range of the original model on the forget classes. In Table 2, we provide the runtime efficiency of our methods, which is similar to relearn time defined in [7] but differs slightly as we report the entire training process rather than just a few epochs needed to achieve the margin.
>
>
> References:
>
> [1] Wu, Jingfeng, et al. "The power and limitation of pretraining-finetuning for linear regression under covariate shift." Advances in Neural Information Processing Systems 35 (2022): 33041-33053.
>
> [2] Ding, Meng, et al. "Understanding Forgetting in Continual Learning with Linear Regression." arXiv preprint arXiv:2405.17583 (2024).
>
> [3] Wu, Jingfeng, et al. "How Many Pretraining Tasks Are Needed for In-Context Learning of Linear Regression?." arXiv preprint arXiv:2310.08391 (2023).
>
> [4] Chen, Xingwu, Lei Zhao, and Difan Zou. "How transformers utilize multi-head attention in in-context learning? a case study on sparse linear regression." arXiv preprint arXiv:2408.04532 (2024).
>
> [5] Fan, Chongyu, et al. "Salun: Empowering machine unlearning via gradient-based weight saliency in both image classification and generation." arXiv preprint arXiv:2310.12508 (2023).
>
> [6] Chundawat, Vikram S., et al. "Zero-shot machine unlearning." IEEE Transactions on Information Forensics and Security 18 (2023): 2345-2354.
>
> [7] A. Golatkar, A. Achille, and S. Soatto, “Eternal sunshine of the spotlessnet: Selective forgetting in deep networks,”.

---

> > ### Author Response · Authors · 2024-11-24
> >
> > Thank you once again for your initial review! We have carefully addressed your comments and provided a detailed response. As the discussion period deadline approaches, we would greatly value your feedback on our response.

---

> > ### Comment · Reviewer_NAAU · 2024-11-29
> > **Response to author replies**
> >
> > I thank the authors for the point-by-point responses, but I am still confused about the following points:
> > 1. Regarding the theoretical analysis, in an overparameterized linear model, it is easy to show that the well-trained model can achieve a 0 loss for all data due to overfitting and then claim the fine-tuning will never work because the remaining data loss has been minimized to 0 even in the original well-trained model. But for other datasets, even for the simplest ones like MNIST, is it possible to achieve an exact 0 training loss? The usage of overparameterized linear regression might not be appropriate for the unlearning analysis.
> >
> > 2. I have reread through the revised Section 5, and I cannot understand the differences between the proposed 'regularizer' and the retaining data loss term in many previous works like Salun, Bad-Teacher and so on. Such papers have tried KL or CE or embedding alignment for unlearning. Thus, I wonder what are the differences between the proposed loss function and the previous ones.
> >
> > 3. Then, regarding the experiments, I understand that this paper does not aim to propose some new benchmarks but focuses on the analysis of why fine-tuning fails. But do the experiments aim to provide some insights about different forms of fine-tuning loss in unlearning? However, without considering the complete SOTA methods, how can we guarantee the experimental results can benefit such methods or future research works?

---

> > > ### Author Response · Authors · 2024-11-29
> > >
> > > Thank you for your continued engagement with our work.
> > >
> > > **Response to Comment 1:**
> > > The well-trained fine-tuned model should achieve 0 loss on the fine-tuned (retaining) data but should not exhibit 0 loss on untrained (forgetting) data, even in overfitting scenarios. However, our empirical results in Table 1 show that naive fine-tuning models fail to effectively forget the forgetting data. This suggests that the fine-tuned model retains information about unseen data from the pretrained model during the naive fine-tuning process. For instance, Table 2 demonstrates that the naive fine-tuning method achieves nearly perfect remaining accuracy (99.76%) on the CIFAR-10 dataset, reinforcing the observation that naive fine-tuning struggles to effectively forget.
> > >
> > > **Response to Comment 2:**
> > > The key difference between our redesigned regularizer and those in previous works lies in the choice of the parameter $\alpha$. As analyzed in the paper, we emphasize that regularization should prioritize retaining accuracy over unlearning accuracy. This principle ensures that the fine-tuning process does not compromise the model’s utility for the remaining data, and we validate this design choice through our experiments.
> > >
> > > **Response to Comment 3:**
> > > As discussed in our previous response, our analysis supports the principle that regularization should prioritize retaining accuracy over unlearning accuracy. The experimental results for ICE-FT benefit from CE-FT (the prior method) by adhering to this principle. For future work, we aim to provide deeper insights into the design of unlearning methods. However, our current focus is on establishing the first theoretical framework and analysis for unlearning.
> > >
> > > Thanks again for your time!

---

> > > ### Author Response · Authors · 2024-12-03
> > >
> > > Dear Reviewer NAAU,
> > >
> > > Thank you for taking the time to review our paper. As the deadline approaches, if we have successfully addressed your concerns, we kindly request you to consider raising your score.
> > >
> > > Best regards,
> > >
> > > The Authors

---

### Official Review · Reviewer_h2Yv · 2024-11-04

**Soundness:** 3
**Presentation:** 2
**Contribution:** 1
**Rating:** 3
**Confidence:** 4

**Summary:**

The paper focuses on the fine-tuning based unlearning scheme, where approximate unlearning is obtained by performing additional learning steps with samples from the retained set to induce "catastrophic forgetting" of the forget set in the model. To understand failure of this technique, the paper considers the linear models and a couple of simplistic data sets where the set of non-zero features of the retain set (i) do not overlap, or (ii) partially overlap with that of the forget set. In both these cases, the theoretical results demonstrate that the fine-tuned based unlearned model (under a specific version of fine-tuning) has very different performance on the forget set compared to the gold standard unlearned model (which is retrained from scratch using only the retain set). Based on these results, the paper discusses a modification to the fine-tuning based unlearning scheme (the paper states them as forms of regularizations) where we are able to reset the parts of the model corresponding to the set of features that are only non-zero on the forget set, and demonstrates how this procedure improves unlearning performance. Based on these insights, the paper motivates the use of a fine-tuning objective for unlearning that combines both unlearning/forget accuracy and accuracy on the retain set, and empirical evaluations highlight how this combined objective improves the unlearning performance of fine-tuning while also maintaining high performance on the retain set.

**Strengths:**

I think one of the main strengths of this paper is the focus on the fine-tuning based unlearning schemes which have (in general) various advantages such as being relatively very efficient, and not requiring the forget set for the unlearning, which has significant practical implications.

**Weaknesses:**

- (W1) To me, one of the main weaknesses of this paper is that there is no clear link between the theory inspired proposed "regularizations" in sections 3 and 4 and the empirical evaluations of section 5. The proposed regularizations (and the related and motivating theoretical analyses) require the knowledge of the distinct and overlapping feature which is usually not available. Thus, it is obvious that these regularization schemes are not practical. However, the connection between the analysis and the use of the combined objective of unlearning/forget accuracy and retain accuracy is not clear at all even in the form of motivation.
- (W2) Even with the considered combined loss function for fine-tuning based unlearning, it is not clear what is novel here. The combined objective has been considered before (as the paper itself mentions), but the paper claims a difference between treating one as an "objective" and one as a "penalty". The difference between what is a penalty term and what is the main objective in equations (6) and (7) seems not compelling enough. Different values of $\alpha$ would lead to different effects, and there is no inherent need to restrict $\alpha \in [0,1]$. Often both can be written as $\lambda (\text{Retain Loss}) + (1 - \lambda) (\text{Forget Loss})$ for some $\lambda \in [0,1]$, and treated as a single hyperparameter which can range from focusing on the loss on the retain set and the performance on forget set. In that respect, to me CE-FT and ICE-FT are the same thing. The main difference between (I)CE-FT and KL-FT is use of KL divergence instead of cross-entropy to penalize the performance on the forget set. In that case, any difference between KL-FT and (I)CE-FT (if there is any) is attributed to the use of KL divergence.
- (W3) This is less of a weakness, but the evaluated method KL-FT and (I)CE-FT require access to the forget set, in contrast to vanilla FT, which does not. This removes the one advantage of FT based fine-tuning. In this case, a more full-scale evaluation across various schemes is warranted including efficient schemes such as Gradient Ascent and influence function based schemes. However, I do not see any novel "method" being presented here (see W2), so there is nothing to evaluate thoroughly here.



Minor comments:
- (C1) In Figure 1, it appears that the quantities plotted are Retain/Unlearning loss (1 - Retain/Unlearning accuracy), which is a bit confusing given that the legend mentions RA/UA, instead of RL/UL as considered in Theorems 1 and 2.
- (C2) In the equation at line 289, there is also a difference of $\mathbf{P} \mathbf{w}\_*^r$ in the definition of $\mathbf{w}\_t$ from the $\mathbf{P}\_r$ in the definition of $\mathbf{w}\_g$ = $\mathbf{P}\_r \mathbf{w}_*^r$. How does that affect the ensuing discussion?
- (C3) In lines 315-319, it is not clear what $d_o$ is as it does not appear to be defined anywhere? Is $d_o$ another name for $d_{\text{lap}}$ or something else?
- (C4) The function $\mathcal{L}_{\text{KL}}(...)$ in equation (6) lacks any clear definition. There are multiple KL divergence based losses discussed in Golatkar et al (2020a), and many of them are related to the Fisher Forgetting scheme proposed therein (unless I have my references wrong). Random label is a baseline there, but I could not find a random label + KL based loss function in there. A more explicit definition here will be very useful.

**Questions:**

- (Q1) Forgetting in linear regression (considered in Sections 3 and 4) would be similar to random data forgetting in classification. What is a "class-wise forgetting" equivalent in the regression setup?
- (Q2) All the evaluations are performed on class-wise forgetting while the theoretical analysis is performed for regression. Is there a reason for why the random forgetting scenario is not considered in the evaluations?
- (Q3) For the results in Table 2, where we have multiple unlearning metrics, how is the hyperparamter $\alpha$ selected for (I)CE-FT and KL-FT?
- (Q4) If hyperparameter optimization is done appropriately (as mentioned above), the main difference between KL-FT and CE-FT from (unmasked) SalUn is the use of KL instead of CE with the forget set. Is there any reason / intuition why we should expect KL divergence based forget set penalty to perform better in terms of all the unlearning metrics compared to the cross-entropy based forget set penalty (that is (I)CE-FT vs KL-FT)?
- (Q5) In the overparameterized regime, the optimal solution to the learning problems (1)-(3) are not necessary singleton sets. Is there any reason we expect the $\arg \min_{\mathbf{w}}$ to be a singleton set and not a set of solutions? If it is in fact not guaranteed to be a singleton set, how does that affect unlearning results in this paper?
- (Q6) In the overlapping feature case, if $d_{\text{lap}} = d$, (that is, full overlap of features) what happens to the bounds? In this case, is there any provable difference between $L(\mathbf{w}_t, D_f)$ and $L(\mathbf{w}_g, D_f)$?

**Details Of Ethics Concerns:**

There are no ethical concerns in my opinion.

---

> ### Author Response · Authors · 2024-11-20
> **Response**
>
> We thank the Reviewer h2Yv for the thorough review and the insightful comments. We hope our following answers can address your concerns.
>
> **Response to Weakness 1:**
> Our primary motivation arises from the empirical observation that fine-tuning can maintain model utility on remaining data but struggles to effectively forget targeted data. Sections 3 and 4 provide theoretical insights to explain and address this issue. Section 5 redesigned a discriminative regularizer aligned with the principle deduced in Section 4: regularization should prioritize remaining accuracy over unlearning accuracy. In response to this weakness, we have rewritten Section 5 to emphasize the redesign rationale for the regularizer and its implications, rather than introducing it. Please check it. Thank you for the comment!
>
> **Response to Weakness 2:**
> We agree that there is limited distinction between our proposed regularization approach and existing objective functions, primarily differing in the choice of the loss function and the parameter $\alpha$. However, the main novelty of our paper lies in providing the first theoretical framework to understand the empirical phenomenon of why naive fine-tuning fails to forget. Based on the analysis in Sections 3 and 4, we aim to address this phenomenon by designing suitable objectives.
> Additionally, Theorem 4.1 provides the rationale for selecting the parameter $\alpha \in(0,1]$ for the unlearning loss, whereas in [1], the constraint $\alpha \in(0,1]$ is applied to the retain loss. **While our objective formulation may appear similar to existing work, our novelty/contribution goes beyond the design of regularize.** In response to this weakness, we have rewritten Section 5 to modify our previous statement. Please check it. Thank you for the comment!
>
> **Response to Weakness 3:**
> We agree with the Reviewer that KL-FT and (I)CE-FT require access to the forget set, unlike Vanilla FT. We appreciate this observation and welcome the opportunity to discuss this point in the context of current machine unlearning settings. Beyond methods requiring access to the forget set, there is also the zero-shot unlearning setting [2], which achieves unlearning without access to either the forget or remaining datasets. Thus, we view this distinction as representing different settings rather than a weakness of our approach.
> Regarding a more comprehensive evaluation, as noted in our response to Weakness 2, the primary novelty of our paper is to provide theoretical analysis and insights into the unlearning process via fine-tuning methods. Thank you for the valuable feedback!
>
> **Response to Minor comments 1:**
> Thank you for pointing this out! The legend in Figure 1 should be RL/UL instead of RA/UA. We corrected this in the revised version.
>
> **Response to Minor comments 2:**
> Thanks for the comment! Our goal is to make the fine-tuned $\mathbf{w}_t$ model as close as possible to the golden model $\mathbf{w}_g$. From the equation on line 289, we can observe the difference between these two solutions, which naturally leads to the following discussion.
>
> **Response to Minor comments 3:**
> Thanks for pointing out this! Yes, $d_o$ refers to $d_{\text {lap }}$ and we corrected the notations in the revised paper.
>
> **Response to Minor comments 4:**
> Thank you for the comment! The function $\\mathcal{L}\_{\mathrm{KL}}(\\cdot)$ in Equation (6) corresponds to the KL divergence between the output of the forgotten model and the one-hot encoded random labels. Specifically, $\\mathcal{L}\_{\\mathrm{KL}}=\\sum\_{i=1}^n \\mathrm{KL} (\\mathbf{w}\_t^{\top}\\mathbf{x}\_i\\| Y\_i^{\prime })$, where $\mathrm{KL}\left(p_i \| q_i\right)=\sum_{j=1}^{|\text{Class}|} p_i(j) \log \left(\frac{p_i(j)}{q_i (j)}\right)$. Regarding the baseline of random label or KL-based loss functions, our primary focus is on providing theoretical analysis to explain the observed empirical phenomenon. Thus, we focus our experiments on naive fine-tuning and the most closely related methods to validate our conclusions. Thank you for your feedback!
>
> **Response to Question 1 and Question 2:**
> Although we use linear regression to analyze the fine-tuning process, our assumptions about the data structure are based on a classification problem, which is why our experiments focus on class-wise forgetting.
>
> **Response to Question 3:**
> In the table 2, we select the corresponding best $\alpha \in (0,1] $ for each methods ((I)CE-FT, KL-FT) to report the results.

---

> > ### Author Response · Authors · 2024-11-20
> > **Continued Response**
> >
> > **Response to Question 4:**
> > Thank you for the question! The primary difference between KL-FT and CE-FT lies in the formulation of the forget dataset penalty. Specifically, KL-FT uses KL divergence: $\mathrm{KL}(p \| q)=\sum_i^{\mid \text {Class } \mid} p(i) \log \frac{p(i)}{q(i)}$, which measures how well the predicted distribution $q$ approximates the true distribution $p$. CE-FT uses cross-entropy: $\mathrm{CE}(p \| q)=-\sum_i^{\mid \text {Class } \mid} p(i) \log q(i)$, which measures the discrepancy between the true labels and predicted probabilities by focusing on the correct class and penalizing incorrect predictions.
> > In our paper, we aim to ensure the fine-tuning process learns an incorrect distribution for the forget dataset, which is why KL divergence is included for completeness. However, whether KL-FT performs better than CE-FT depends on the dataset, task, and evaluation metrics. In practice, their performance may vary depending on hyperparameter tuning and the structure of the forget set. We will further investigate this distinction in future work. Thank you for raising this insightful question!
> >
> > **Response to Question 5:**
> > Thank you for the question! In our paper, we consider that, in the overparameterized setting, the solutions for the MSL loss (line 161) are not singleton sets. However, among these solutions, we focus on the one with the smallest $\ell_2$-norm of the parameter change, as specified in Eqs. (1), (2), and (3). Consequently, the learning problems (1)-(3) represent unique solutions.
> >
> > **Response to Question 6:**
> > Thanks for bring this point to us! If $d\_\text{lap} = d$, then we have $L(\mathbf{w}\_t, D\_f) = 0$ and  $L (\\mathbf{w}\_g, D\_f) = \\| ( \\mathbf{P}\_r -\\mathbf{I} ) \\mathbf{w}\_*\\|\_{\\frac{1}{n\_f} \\mathbf{X}\_f\\mathbf{X}\_f^{\\top}}.$
> > It can be observed that a gap still exists between the golden model and the fine-tuned model, even when we assume a full overlap of features. This gap arises due to the difference between the golden model solution $\\mathbf{P}\_r \\mathbf{w}\_*^{}$ and real solution $\\mathbf{w}\_*^{}$.
> >
> > Reference:
> >
> > [1] Fan, Chongyu, et al. "Salun: Empowering machine unlearning via gradient-based weight saliency in both image classification and generation." arXiv preprint arXiv:2310.12508 (2023).
> >
> > [2] Chundawat, Vikram S., et al. "Zero-shot machine unlearning." IEEE Transactions on Information Forensics and Security 18 (2023): 2345-2354.

---

> > > ### Author Response · Authors · 2024-11-24
> > >
> > > Thank you once again for your initial review! We have carefully addressed your comments and provided a detailed response. As the discussion period deadline approaches, we would greatly value your feedback on our response.

---

> ### Comment · Reviewer_h2Yv · 2024-11-26
>
> I thank the authors for the updated manuscript and the point-by-point responses. Based on reading the revised manuscript, I continue to struggle with some of the core conceptual aspects of the paper:
>
> It is a bit counterintuitive that the new "regularization" is "prioritizing retaining accuracy over unlearning accuracy" in fine-tuning based unlearning. To the best of my understanding, this is something that already happens in fine-tuning as fine-tuning literally just optimizes for the retaining accuracy with no regards to the unlearning accuracy (as also defined in (2) except for the $\mathbf{w}_o$ term in the loss $|| \mathbf{w} - \mathbf{w}_0 ||^2$). So fine-tuning is already only prioritizing "retaining accuracy" in the most extreme sense, and still is unsuccessful in unlearning (as shown in section 3).
>
> Thus, the main message I take away from the (re)reading this paper is the following:
> - **M1:** (section 3) Fine-tuning (as defined in (2)) is not successful in unlearning as there is a significant gap between the golden model $\mathbf{w}_g$ and the fine-tuned model $\mathbf{w}_t$ (although there are some caveats; see **Note A** and **Note B**).
> - **M2:** (section 4) In the overlapping setup, it is better to leave the overlapping feature weights (instead of resetting them) as this provides the best tradeoff between unlearning loss and retaining loss among all two ways to "regularize" (reset) the weights for fine-tuning. However, in the (pratically usual) case where $d_{\text{lap}} = d$, this regularization is a no-op as there is nothing to be done before fine-tuning, and we are back to basic fine-tuning (except that we are solving a specific instance of fine-tuning defined in (2)). So effectively, the best regularization is to just do fine-tuning.
> - **M3:** (section 5) This message is reiterated with the experiments where the authors show that putting more weight on the retaining accuracy is better for unlearning/utility tradeoff. However, again, note that putting all the weight on the retain accuracy is the usual fine-tuning based unlearning.
>
>
> **Note A:** The $d_{\text{lap}} = d$ case is not clear. Given $\mathbf{w}\_\* = \mathbf{w}\_\*^{\text{lap}}$ when $d_{\text{lap}} = d$, let the original problem (1) solution be $\mathbf{w}\_o = \mathbf{w}\_\*$. Since $y\_r = X\_r^\top \mathbf{w}\_\*$, $\mathbf{w}\_t = \mathbf{w}\_\*$ with $L(\mathbf{w}\_t, D\_r) = 0$ since this value would minimize $||\mathbf{w}\_o - \mathbf{w} ||^2$. Also $L(\mathbf{w}\_t, D\_f) = 0$ so no forgetting happens. However, $\mathbf{w}\_g = \mathbf{w}\_\*^{\text{lap}} = \mathbf{w}\_\* = \mathbf{w}\_t$. So $L(\mathbf{w}\_g, D\_r) = 0$ but also $L(\mathbf{w}\_g, D\_f) = L(\mathbf{w}\_*, D\_f) = L(\mathbf{w}\_t, D\_f) = 0$, so the golden unlearned model also has low unlearning loss and no gap exists between the golden and the fine-tuned model. So it is not clear when and where the mentioned gap would exist.
>
> **Note B:** It seems to me that the results in section 3 regarding the gap between $L(\mathbf{w}\_t, D\_f)$ and $L(\mathbf{w}\_g, D\_f)$ stems from the formulation of the objective $\arg\min || \mathbf{w} - \mathbf{w}\_o ||^2$ for fine-tuning in (2) involving the original training solution $\mathbf{w}\_o$. This makes sense given that the fine-tuning process initiates from $\mathbf{w}\_o$. However, this formulation seems critical to have the gap between fine-tuning and golden model. Since the golden model minimizes the norm $|| \mathbf{w} ||^2$ in (3), it drops the disentangled $\mathbf{w}\_\*^f$ part of the weights.  In contrast, the fine-tuning solution in (2) is limited to be a minimal distance solution with the $|| \mathbf{w} - \mathbf{w}\_o ||^2$ term, thereby forcing the $\mathbf{w}\_t$ to keep the $\mathbf{w}\_\*^f$ component of the weights. To me, this is not an inherent behaviour of fine-tuning based unlearning, but rather an artifact of the fine-tuning problem formulation. Instead if the fine-tuning problem (2) was formulated as following where we again look for the lowest-norm solution while being close to the initial $\mathbf{w}\_o$, it is not clear if the results would continue to hold
>
> $$ \arg \min_{\mathbf{w}} || \mathbf{w} ||^2 \text{ s. t. } \text{(i)} y_t = X_t^\top \mathbf{w}, \text{(ii)} || \mathbf{w} - \mathbf{w}_o ||^2 \leq \epsilon.$$
>
> In this case, the preservation of the $\mathbf{w}\_\*^f$ component in the $\mathbf{w}\_t$ is no longer ensured, and the unlearning loss gap between $L(\mathbf{w}\_t, D\_f)$ and $L(\mathbf{w}\_g, D\_f)$ would probably depend on $\epsilon$ in constraint (ii), which corresponds to how thorough we allow the fine-tuning (using the retain loss) to be. If $\epsilon$ is large enough that $|| \mathbf{w}_o - \mathbf{w}_g ||^2 \leq \epsilon$, then fine-tuning should find $\mathbf{w}\_g$.

---

> ### Comment · Reviewer_h2Yv · 2024-11-26
>
> Minor: Given a ground-truth distribution $p$ and a predicted distribution $q$ (which we are optimizing over), is it not the case that minimizing the cross-entropy loss
> $$\min_q CE(p | q) = \min_q  -\sum_i  p(i) \log q(i)$$
> is equivalent to minimizing the KL-divergence
>
> \begin{align*}
> \min_q KL(p|q) & = \min_q \sum_i p(i) \log \frac{p(i)}{q(i)} \newline
>      & = \min_q \sum_i \left[ p(i) \log p(i) -  p(i) \log q(i) \right] \newline
>      & \equiv \min_q - \sum_i p(i) \log q(i)
> \end{align*}
>
> What am I missing here if they are equivalent optimization problems? Is the actual difference that we use different ground-truth distributions $p$ for CE vs KL?

---

> ### Author Response · Authors · 2024-11-26
>
> We sincerely thank the reviewer for the thoughtful feedback and for the opportunity to clarify some core aspects of our paper! We address each of the points raised below to enhance the clarity and understanding of our work.
>
> **Response to Counterintuitive Nature of "RA over UA":**
>
> We understand that it might seem counterintuitive that the regularization involves "prioritizing retaining accuracy over unlearning accuracy" in fine-tuning-based unlearning. In standard fine-tuning methods, the optimization focuses solely on retaining accuracy (RA) without explicitly considering unlearning accuracy (UA). This lack of an unlearning objective is precisely why naive fine-tuning often fails to effectively unlearn.
> Our method revisited a regularization term that explicitly balances RA and UA. By incorporating both objectives into the fine-tuning process, we can improve unlearning performance while maintaining acceptable performance on the retained data. This regularization is not inherent in standard fine-tuning, which does not address UA at all. Therefore, the discussion on regularizer differs from naive fine-tuning by directly incorporating unlearning into the optimization objective.
>
> **Response to M1: Gap Between Fine-Tuned Model and Golden Model:**
>
> In Section 3, we analyze scenarios where fine-tuning, as defined in equation (2), does not achieve the same unlearning performance as the golden model. We acknowledge that when the overlapping dimension $d_{\text {lap }}=d$, the situation requires further examination.
>
> **Addressing Note A**:
>
> When $d = d_{lap}$, the solutions for each problem should be $\\mathbf{w}\_o^{} = \\mathbf{P}\\mathbf{w}\_*^{}$, $\\mathbf{w}\_g^{} = \\mathbf{P}\_r\mathbf{w}\_*^{}$, $\\mathbf{w}\_t^{} = \\mathbf{P}\\mathbf{w}\_*^{}$. Therefore, the loss for Note A should be $$L(\\mathbf{w}\_g^{}, D\_f) = \\frac{1}{n\_f}\\|\\mathbf{X}\_f (\\mathbf{P}\_r - \\mathbf{I})\\mathbf{w}\_*^{}\\|^2 \\neq L(\\mathbf{w}\_t^{}, D\_f) = 0.$$
> There is still a gap between fine-tuned model $\mathbf{w}_t$ and $\mathbf{w}_g$. This demonstrates that even when all features overlap, the fine-tuned model does not match the golden model's unlearning performance.
>
> **Addressing Note B**:
>
> The reviewer suggests that the gap arises due to the specific formulation of the fine-tuning objective, particularly the $\\|\mathbf{w}-\mathbf{w}_o\\|^2$ term in equation (2). In the following, we show that when considering the fine-tuning process as $$
> \arg \min\_{\mathbf{w}} \\|\mathbf{w}\\|^2 \text { s. t. (i) } \mathbf{y}_t=\mathbf{X}_t^{\top} w, \text { (ii) }\\|\mathbf{w}-\mathbf{w}_o\\|^2 \leq \epsilon
> $$
> There are still gaps between fine-tuned model $\mathbf{w}_t$ and golden model $\mathbf{w}_g$. The results can differ based on whether the constraint is active or inactive:
>
> **1) The Inequality Constraint is Inactive.**
>
> Then the fine-tuned model for distinct case (overlapping case can be derived similarly) is $\mathbf{w}_t = \mathbf{P}_t \mathbf{w}_r^* $ and the results are $L(\mathbf{w}_t, D_f) = \frac{1}{n_f}\\|\mathbf{X}_f\mathbf{w}_f^*\\|^2 = L(\mathbf{w}_g, D_f)$, and $L(\mathbf{w}_t, D_r) = \frac{1}{n_r}\\|\mathbf{X}_r(\mathbf{P}_t -\mathbf{I})\mathbf{w}_r^*\\|^2 \neq L(\mathbf{w}_g, D_r)$. In this case, fine-tuned model $\mathbf{w}_t$ indeed can achieve same unlearning performance as golden model $\mathbf{w}_g$, while there is a gap on the retaining performance.
>
> **2) The Inequality Constraint is active.**
>
> If the solution exists, then active inequality constraint $\\|\mathbf{w}_t-\mathbf{w}_o\\|^2=\epsilon$ holds. The results become:
> $$L(\\mathbf{w}\_t, D\_f)=  \\frac{\\epsilon}{n\_f}\\|\\mathbf{X}\_f\\|^2 , \\quad \\text{and} \\quad L(\\mathbf{w}\_t, D\_r)=  \\frac{\\epsilon}{n\_r}\\|\\mathbf{X}\_r\\|^2 .$$
>
> In this case, it can be observed that regardless of whether the features are distinct or overlapping a consistent gap related to $\epsilon$ exists between the fine-tuned model and the golden model. When $\epsilon=0$, the scenario reduces to our analysis, demonstrating that similar results arise under this problem formulation. Thanks for the insightful comment, we will include this discussion in our revised version.

---

> > ### Author Response · Authors · 2024-11-26
> >
> > **Response to M2:**
> >
> > We respectfully disagree with the assertion that our results are an artifact of our problem formulation. Our study focuses on class-wise unlearning with distinct features, a common scenario in classification tasks where different classes possess unique attributes. In the following, we provide practical relevance of distinct features:
> >
> > **Example with CIFAR-10 Dataset:**
> >
> > Car Class: Features like wheels and headlights.
> >
> > Bird Class: Features like wings and feathers.
> >
> > In such datasets, the features associated with different classes are distinct. Our analysis is designed to reflect this practical aspect, where unlearning a class involves removing specific, non-overlapping features. This justifies our focus on cases where the forget and retain datasets have distinct features.
> >
> > **Regarding Full Feature Overlap:**
> >
> > We acknowledge that when $d=d_{\text {lap }}$, the forget and retain datasets share all features, however, it resembles random data unlearning. This scenario is not the primary focus of our work, as it does not align with the classwise unlearning framework we investigate. Our primary interest lies in situations where classes can be distinguished by their unique features a common occurrence in real-world applications.
> >
> > **Response to M3:**
> >
> > We acknowledge that setting the weight parameter $\alpha=0$ in our objective function effectively reduces our method to naive fine-tuning, focusing solely on retaining accuracy without explicitly addressing unlearning. However, our analysis and experimental results aim to provide more validation for our previous analysis. Our experiments demonstrate that placing more weight on retaining accuracy indeed improves performance on the forget dataset without compromising retaining effectiveness. While naive fine-tuning serves as a baseline case, it can not provide insights into the trade-off between unlearning and retaining with only one objective.
> >
> > **Response to Minor:**
> >
> > Thank you for your insightful comment. You are correct that, theoretically, minimizing the crossentropy loss $\text{CE}(p \mid q)$ over $q$ is equivalent to minimizing the KL-divergence $\text{KL}(p \mid q)$ over $q$, since the entropy term is constant with respect to $q$.
> >
> > However, in our implementation, we observed practical differences due to how these loss functions are handled in machine learning libraries. Specifically, cross-entropy loss functions typically expect logits (unnormalized scores) and internally apply softmax, while KL-divergence functions often require probabilities or log probabilities as inputs. Additionally, some implementations compute $\text{KL}(q \mid p)$ instead of $\text{KL}(p \mid q)$, affecting the optimization direction.
> >
> > These implementation details can lead to different optimization behaviors, even when using the same ground-truth distribution $p$. Therefore, while the theoretical objectives are equivalent, practical differences arise due to the specifics of function implementations and input requirements.
> >
> > We appreciate your observation and will clarify this point in our revised manuscript to better reflect the theoretical equivalence and practical considerations!
> >
> > Thank you again for your continued engagement with our work and the thoughtful feedback you've provided!

---

> > > ### Comment · Reviewer_h2Yv · 2024-11-28
> > >
> > > I thank the author for promptly analyzing the new fine-tuning objective and for responding to all my comments.
> > >
> > > **Regarding gap between $ L( \mathbf{w}_g, D_f ) $ and $ L( \mathbf{w}_t, D_f ) $ in Addressing Note A**
> > >
> > > Thanks for the clarification. There would be a significant gap in the $d_{\text{lap}} = d$ case only if $ \mathbf{P} $ and $ \mathbf{P}_r $ are significantly different, which is straightforward with non-overlapping features, but the existence of such a difference is a bit more non-trivial with overlapping features (especially when all features are overlapping). If $ \mathbf{ P } \approx \mathbf{ P }_r $, then the gap might not be statistically significant.
> > >
> > > **Regarding Addressing Note B with distinct features**
> > >
> > > I have not verified the derivation, but assuming that they are correct, one would make the following conclusions:
> > > - In the case where the constraint (ii) is inactive, the forgetting quality of fine-tuning matches the forgetting quality of the golden model. This is the opposite conclusion to the one in Theorem 3.2.
> > > - There is a difference in the retaining accuracy of the fine-tuned and golden model (again counter to Theorem 3.2, where we see that the RL matches between fine-tuning and golden). However, this relies on the difference between the projection $\mathbf{P}_t$ of the fine-tuning set $D_t$ and the projection $\mathbf{P}_r$ of the full retained set $D_r$ where $D_t \subset D_r$. Assuming that the data is i.i.d. (which is standard), unless $D_t$ is dramatically smaller than $D_r$ (or adversarially sampled), the $\mathbf{P}_t$ and $\mathbf{P}_r$ should not be that different, and thus the gap in retained accuracy should not be that large.
> > > - When the constraint (ii) is active, it is expected that there would be a gap as I had mentioned although I would want to understand derivations of the posted results.
> > >
> > > So, with this formulation for fine-tuning (different than the one in (2)), it seems that fine-tuning is statistically similar to the golden model both in forgetting and in retained accuracy as long as
> > > - (i) the projection $\mathbf{P}_t$ of the subset $D_t$ is not significantly different than the projection $\mathbf{P}_r$ of the full set $D_t$ (which should be the case with large enough fine-tuning set), and
> > > - (ii) we allow the fine-tuning to be sufficiently thorough (that is, we allow $\epsilon$ to be large around).
> > >
> > > To me, this is a very different conclusion showing that fine-tuning **can work** if we solve the right fine-tuning problem without any need for having unlearning accuracy as a regularization. This conclusion is very different from the conclusion drawn regarding fine-tuning as defined in (2), which is then used by the authors to motivate the use of unlearning accuracy in the fine-tuning process. This is my main reason for saying that the formulation of the fine-tuning problem significantly affects the conclusions we are drawing here, which makes the overall theoretical message unclear.
> > >
> > > Of course, practically utilizing some form of unlearning loss (alongside loss on the retained set) during unlearning has been seen to be extremely useful. However, if one formulation of fine-tuning tells us that we need to include some unlearning loss, while a slight (but significant) modification of the fine-tuning objective tells us that fine-tuning matches golden if some conditions (independent of any unlearning loss) are satisfied, then it tells me that there is something in this theoretical framework we are not properly understanding.

---

> > > > ### Comment · Reviewer_h2Yv · 2024-11-28
> > > >
> > > > **Regarding the CIFAR-10 example and the connection between Full Feature Overlap and random unlearning**
> > > >
> > > > I am not disagreeing that different classes can have different set of **implicit** features. But the model we are trying to unlearn with do not unfortunately have access to these nicely disentagled features that are directly tied to the output classes. If it were the case, both learning, and unlearning would be easy, no fine-tuning needed. For example, in the case of Section 4, scenario 1, once we reset the weights corresponding to $d_f$, we are done. I do not think we even have to do the fine-tuning since this weight is already the golden model. So even in the class forgetting scenario, there is at least full overlap in the "explicit features". We can claim that we can use some form of disentangled representation learning, and this model being analysed works on top of that, and that is fine. But then there is the case of unlearning that representation model but that is beyond the scope of what is being studied in this paper. But this is a relatively minor point, and I appreciate the explanations provided by the authors.
> > > >
> > > >
> > > > **Regarding difference (or lack thereof) between KL divergence and CE as utilized in this work**
> > > >
> > > > I would be quite surprised if the "practical implementation differences" between the two equivalent objectives would lead to statistically significant differences. The exact numbers can be different, but it would be very surprising if those lead to marked variations. I would actually be very concerned if the differences are big since that would imply wrong implementations in these machine learning libraries. However, much has been said in this submission regarding the difference between (I)CE-FT vs KL-FT (the 3 out of the 4 methods evaluated), whereas, theoretically they are all the same objectives upto (i) the choice of the hyperparameters, and (ii) the practical implementation differences of CE and KL for the same ground-truth distribution $p$ and predicted distribution $q$.

---

> > > > > ### Author Response · Authors · 2024-12-03
> > > > >
> > > > > Dear Reviewer h2Yv,
> > > > >
> > > > > Thank you for taking the time to review our paper. As the deadline approaches, if we have successfully addressed your concerns, we kindly request you to consider raising your score.
> > > > >
> > > > > Best regards,
> > > > > The Authors

---

> ### Author Response · Authors · 2024-11-29
>
> We thank the reviewer once again for the thoughtful responses and insightful comments.
>
> **Response to Main Concern:**
>
> We understand and appreciate the reviewer's primary concern regarding the differing conclusions under different conditions, specifically:
> 1. When the fine-tuning dataset $D_t$ is similar to $D_r$, and the inequality constraint becomes inactive.
> 2. When the inequality constraint is active, and $\epsilon$ is sufficiently large.
>
> For the first scenario, if the inequality constraint is inactive, the fine-tuned model $\mathbf{w}_t$ can achieve perfect unlearning/retention loss, same as the golden model $\mathbf{w}_g$. However, in what situations does the inequality become inactive? If this occurs, the problem becomes disconnected from the pretrained model $\mathbf{w}_o$, meaning the fine-tuned model cannot truly be regarded as a fine-tuned model since it does not utilize the pretrained model's information.
>
> For the second scenario, if $\epsilon$ is large, it suggests that the pretrained model does not significantly contribute to the fine-tuned model. Consequently, both the remaining loss and unlearning loss deviate from the golden model and produce suboptimal results.
>
> Regarding Section 4, the elimination of forgetting data features assumes that the structure of the data/model is already known, as noted in Minor 1. This assumption is indeed restrictive in practice. To address this, we propose adding another unlearning loss function to the naive fine-tuning approach, which aligns with the overarching idea of eliminating features associated with the forgetting data.
>
> We are grateful for the reviewer's insightful comments and will incorporate these discussions into our revised version. However, we still believe the current problem setup is the most suitable way to describe the unlearning process compared to the scenarios discussed above.
>
> **Response to Minor 1:**
>
> We agree with the reviewer that accessing nicely disentangled features is not always feasible. For analytical purposes, we started with a simplified case but are keen to extend this framework to deeper analyses with more generalized assumptions and models. There is substantial existing work on feature learning that describes similar challenges, such as [1], and we aspire to extend our work along these lines.
>
> Reference:
> Allen-Zhu, Zeyuan, and Yuanzhi Li. "Towards understanding ensemble, knowledge distillation and self-distillation in deep learning." arXiv preprint arXiv:2012.09816 (2020).
>
> **Response to Minor 2:**
>
> Thank you for the reviewer's feedback. During our experiments, we observed that the choice of function (CE or KL) leads to only minor differences in the outcome. The more significant factor lies in selecting the appropriate parameter.  That's why we follow your suggestion to rewrite the Section 5 part. Thanks again for your suggestion!

---

### Official Review · Reviewer_bHDY · 2024-11-04

**Soundness:** 3
**Presentation:** 4
**Contribution:** 3
**Rating:** 6
**Confidence:** 3

**Summary:**

This work explores a known issue with naïve fine-tuning approaches for the machine unlearning problem: it struggles to forget the targeted data. To address this problem, the authors begin by constructing a synthetic experiment (an overparameterized linear regression model) and show that, in this case, fine-tuned weights decompose into two components—one that targets the remaining data and one that can be considered the residual from the data to forget.

Based on this decomposition, they compare two approaches that aim to reduce the unwanted component in the final solution. They find both empirically and theoretically that the approach focusing on solving for the remaining data, rather than solely forgetting the target data, performs better. This observation leads them to propose a loss function that prioritizes overall accuracy over the forgetting term. The performance of this loss is empirically evaluated on a real dataset.

**Strengths:**

1. The overall paper is well-structured and pleasant to read.

2. The theoretical results inspired insights into the practical effect of the real loss, which were validated on a real data example. The theoretical section is strong and insightful. Supporting empirical experiments presented alongside each step to further ground the understanding are sensible, well presented, and convey each point effectively.

**Weaknesses:**

1. I found the presentation of the two terms in the loss (Eq. 6), with one being the main term and the other the “regularizing term,” to be problematic. Unless I am mistaken, this distinction is artificially created by placing an arbitrary cap on the regularization scaling term $\alpha \in 0-1$, which effectively upper bounds the contribution of each term to the loss. Therefore, the second point, 2) Regularization Focus, does not represent a real difference, in my opinion, and needlessly obfuscates the work (though I appreciate the desire to differentiate from previous work). The crux of the theoretical insight is that the accuracy term should be given more importance than the unlearning term. What is the procedure to tune this $\alpha$ parameter, as it is a crucial value of the experiment? This detail is missing in the main text and should definitely be included.
From my understanding, the ICE-FT is effectively the same as CE-FT with $\alpha \in [1,\infty]$ instead of being restricted to $\alpha \in 0,1$. (If that is not the case, please add an explicit formula for the ICE-FT and disregard the following comment). Related to that point, the optimal $\alpha$  value for CE-FT should then be then as close as 1 as possible. Is it the case? In figure 4b, why is ICE and CE with alpha=1 not equal?

2. The experiment on the real datasets could be improved in a few areas. First, why is only one fine-tuning baseline presented? Second, details about the tuning and training procedure are missing, which are particularly important as the authors highlighted the sensitivity to the $\alpha$ parameter, and there is no single target metric to optimize for. The presentation of the results on the real datasets could also be improved. Looking at Table 2, it is hard to see how each method performs. Showing Figure 3 for all the datasets could help for example, or using UA vs. RA curves would be more convincing than presenting point predictions in a table. Additionally, the legend in Figure 4b is hidden behind the lines and has the wrong colors.

3. Assumption 3.1 and Remark 1. I found the construction of the matrices $F$ and $R$ to be a little bit vague. Shouldn’t it have some constraints on$d_f$ and $d_r$ since they $w_{f*}$ and $w_{r*}$ are to be exact solutions to both problems? The remark really feels more like a part of the assumption (similar comments apply to Remark 2 paired with Assumption 3.3). The concept of feature overlapping should be clarified, as it has a different meaning than how it is usually used.
These points should be clarified to understand the limits of the conclusions we can draw from this synthetic setup.



Minor
- The discussion after Theorem 3.2 could be clarified. The constructed example is a scenario where the weights learned from the different tasks are completely orthogonal to each other, so the fine-tuning step is performed in a totally unrelated space. This is a great illustrative example to showcase how fine-tuning cannot affect the performance on the initial task. However, this is very specific to this particular crafted setting with extreme overparameterization. Therefore, it doesn’t really “suggest that the fine-tuning model is unable to forget the information it previously acquired from…” in general; it applies only to that particular model. Discussing the relations to the setup from Ding et al. (2024) would be interesting.

- The discussion following Theorem 3.2 and Theorem 3.4 feels somewhat repetitive, as the same points are made. You could instead focus more on discussing the differences between the two.
- There is no reference in the main text to the appendix.
- The norm in Eqs. 1, 2, and 3 is undefined.
- The point that "we favor the principle that regularization should prioritize remaining accuracy over unlearning accuracy" should be made before presenting the loss. Without it, the loss feels somewhat disconnected from the previous section.
- Consider introducing UA and RA earlier (perhaps as part of the problem description), as you present various results before their formal introduction.

Typos and small details/suggestions.

- (13), (38) (109), and a few others… typo inverted bracket ‘removing’ -> `removing’.
- Table 1:  FT (Fine-Tuning) Methods. -> FT (Fine-Tuning) Method ?
- In Section 2, overparameterized linear regression should be defined when it is introduced (n<<d). (also typo overparamterized)
- You could define RL and UL with mathematical notation as they are introduced.
- Theorem 3.2 , missing reference to the proof in the appendix after the  theorem statement.
- Font of Figure 1 are too small.
- Introduce the notation $y’$ as a wrong label to before Eqn. 6.
- You can drop the line ``The evaluation metrics include Unlearning Accuracy (UA), MIA-Efficacy, Retaining Accuracy (RA), Test Accuracy (TA), and RunTime'' in Table 2 caption to save space.

**Questions:**

1. For Figure 2a and the accompanying discussion and conclusions, it is important to note the fraction of overlapping features, as it likely has a significant impact on these aspects. Could you comment on this point?

2. Line 413: “Notably, the regularization parameter is typically constrained to the range (0, 1].” What is notable about that? Could you clarify the point of this comment? I couldn’t find any reference in (Fan et al., 2023) to bounding this parameter to that range.

3. Why is UA–RA more important than RA–TA? UA seems to relate to the training set.

---

> ### Author Response · Authors · 2024-11-20
> **Response**
>
> We are grateful to Reviewer bHDY for the detailed review and constructive feedback. We hope address your concerns accordingly.
>
> **Response to Weakness1:**
> - (1) Setting of Regularization: You raised an important point regarding the scale of the two terms in the loss (Eq. 6). Our design intentionally restricts $\alpha \in (0,1]$ to prioritize retaining accuracy over unlearning accuracy, which differs from the approach in [1], where unlearning accuracy is considered more critical. We understand this distinction may appear to be an arbitrary design choice; however, it aligns with the goals of our previous conclusion: we should upperbound the unlearning accuracy instead of remaining accuracy in [1].
> - (2) Experimental Setting of $\alpha$: In our experiment, we explore the sensitivity of the regularization parameter within the range $[0,0.1, \ldots, 0.9]$. Yes, ICE-FT is exactly the same as CE-FT when $\alpha \in[1, \infty]$. However, as shown in Figure $4b$, increasing $\alpha$ did not improve the performance of unlearning accuracy, and the retain accuracy remained unchanged.
> - (3) Difference Between ICE and CE Under $\alpha$: The optimal $\alpha$ for CE-FT should ideally be greater than 1; however, the analysis in [1] restricted the range of $\alpha$, making 1 the best value under this limitation. Furthermore, in Figure $4b$, ICE and CE converge and are identical when $\alpha=1$. To highlight the differences between the two methods, Figure $4 b$ only presents the results for $\alpha$ in the range $[0.1,..., 0.9]$.
>
> **Response to Weakness2:**
> - (1) Why is only one fine-tuning baseline presented? Our primary motivation stems from the empirical observation that while fine-tuning can maintain the utility of the model on the remaining data, it struggles to effectively forget the targeted data. We aim to provide theoretical analysis to explain this phenomenon. At this stage, we focus on fine-tuning-based experiments to validate our initial analysis. In the future, we plan to extend our work by including additional analysis and experiments with other unlearning methods, such as gradient ascent.
> - (2) Additional experimental details: We have included details about tuning and training settings, along with more experimental results, in Appendix A.2. Specifically, we provide information on the unlearning setup, datasets, model architectures, and the loss values for different $\alpha$ values in ICE-FT and CE-FT.
> - (3) Improving experimental presentation: Thank you for your suggestions on improving the experimental presentation! We have addressed this by: (a) Extending Figure 3-style visualizations to all datasets. (b) Including UA vs. RA curves for CIFAR-10 in Figure 4, with the Retain curve (blue line) and Forget curve (orange line) highlighted. We are working on further extending these visualizations to other datasets. (c) Could you clarify your concerns about Table 2’s organization and the incorrect legend colors in Figure 4b? This feedback would help us improve these aspects. Thank you!
>
> **Response to W3:** We use the matrices $\\mathbf{F}$ and $\\mathbf{R}$ to illustrate the structure of the data matrix. Specifically, the entire data matrix can be represented as $\\mathbf{X}=[\\mathbf{X}\_r, \\mathbf{X}\_f]$, where $\\mathbf{X}\_r^{\\top}=[\\mathbf{R}^{\top}, \\mathbf{0}]$ and $\\mathbf{X}\_f^{\\top}=[\\mathbf{0}, \\mathbf{F}^{\top}]$.
> Consequently, based on the problem setup (lines 141-157), we can deduce the existence of $\\mathbf{w}\_*^f$ and $\\mathbf{w}\_*^r$ such that $\\mathbf{w}\_*^{} =\\mathbf{w}\_*^r + \\mathbf{w}\_*^f$, $\\mathbf{y}^f=\\mathbf{X}\_f^{\top} \\mathbf{w}\_*^f$ and $\\mathbf{y}^r=\\mathbf{X}\_r^{\top} \\mathbf{w}\_*^r .$
> The dimensions $d\_f$ and $d\_r$ should be consistent with the partitioning of the feature space. The term "feature overlapping" in our work refers to scenarios where some features are shared between $\\mathbf{X}\_r$ and $\\mathbf{X}\_f$.

---

> > ### Author Response · Authors · 2024-11-20
> > **Response**
> >
> > **Response to Minor 1:**
> > Thank you for your insightful comment! Yes, in this work, we start with the overparameterized linear regression model and aim to extend the analysis to more general cases in the future. We will revise our statements to clarify that the conclusions are restricted to our specific case. Additionally, we provide more discussion on the relationship between our setup and Ding's work here. Both studies are based on the overparameterized linear regression model. However, the key difference is that our case involves two tasks: the first task is pretraining on all the data, and the second task is fine-tuning on a subset of the data. In contrast, Ding's work focuses on the continual learning setting, where each task has its own independent dataset.
> >
> > **Response to Minor 2:**
> > Thank you for the suggestion. Indeed, the overlapping case can be seen as a generalization of the distinct case. The discussions following Theorems 3.2 and 3.4 aim to highlight the fact that naive fine-tuning fails to achieve effective unlearning. In the subsequent sections, we shift our focus to the differences between the two cases.
> > We will revise the discussion to reduce redundancy and place greater emphasis on how the two cases are related and how they differ. Thanks!
> >
> > **Response to Minor 3:**
> > Thank you for pointing this out! We revised the paper to include clear references to the appendix in the main text.
> >
> > **Response to Minor 4:**
> > Thank you! It should be the $\\|\cdot\\|\_2$ norm. We corrected this in the revision.
> >
> > **Response to Minor 5:**
> > Thank you for the suggestion! We rewrote Section 5 to better connect to the previous section, please check it.
> >
> > **Response to Minor 6:**
> > Thank you for the suggestion! We introduced UA(UL) and RA(RL) earlier in the revised version, as part of the problem description.
> >
> > **Response to Typos:** We sincerely thank Reviewer bHDY for the detailed suggestions! We have addressed these in our revised version as follows: (a) Fix typos like inverted brackets in (13), (38), and (109), as well as "removing" $\rightarrow$ " removing " and "overparamterized" $\rightarrow$ "overparameterized."
> > (b)  Update Table 1 header to "FT (Fine-Tuning) Method."
> > (d) Define overparameterized linear regression $(n \ll d)$ when it is introduced in Section 2.
> > (e) Introduce mathematical notation for RL and UL when they are first defined.
> > (f) Add a reference to the proof in the appendix after Theorem 3.2.
> > (g) Adjust the font size in Figure 1 to improve readability.
> > (h) Introduce the notation for a wrong label before Eq. (6).
> > (i) Shorten the caption for Table 2 by removing the line about evaluation metrics to save space.
> >
> > **Response to Question 1:** Thank you for the question! We also observe a tendency for the unlearning loss to decrease as the fraction of overlapping features increases. This is an intuitive conclusion, as overlapping features likely play a key role in retaining shared information. However, under the current problem setup, we cannot directly prove this observation, which is why it is not discussed in detail in the paper. We believe this phenomenon may be influenced by other factors, such as the importance of the overlapping features. Exploring how to quantify these factors and their impact on unlearning dynamics will be a focus of future work. Thank you for your question!
> >
> > **Response to Question 2:**
> > Thank you for the question! Based on our analysis, we favor the principle that regularization should prioritize remaining accuracy over unlearning accuracy, which supports constraining $\alpha \in$ $(0,1]$ for the unlearning term. While this specific range may not be explicitly mentioned in [1], we reviewed their code and directly consulted the authors, who confirmed that $\alpha \in(0,1]$ is the range they used in their implementation.
> >
> > **Response to Question 3:**
> > Thank you for the question! There are many evaluation metrics in the unlearning problem. However, we prioritize the unlearned model's behavior to closely approximate the retrained (golden) model. Therefore, the metrics we consider are not limited to UA, RA, and TA.
> >
> > Thank you again for your careful review and helpful feedback!
> >
> > Reference:
> >
> > [1] Chongyu Fan, Jiancheng Liu, Yihua Zhang, Dennis Wei, Eric Wong, and Sijia Liu. Salun: Empowering machine unlearning via gradient-based weight saliency in both image classification and generation. arXiv preprint arXiv:2310.12508, 2023.

---

> > > ### Author Response · Authors · 2024-11-24
> > >
> > > Thank you once again for your initial review! We have carefully addressed your comments and provided a detailed response. As the discussion period deadline approaches, we would greatly value your feedback on our response.

---

> ### Comment · Reviewer_bHDY · 2024-11-26
>
> Thank you for your response.
> ### W1 / W2.2
> I still have reservations about this point, along with the fact that it remains unclear how the $\alpha$ value is set in your experiment. I couldn't find in Appendix A.2 how the $\alpha$ value is set?
>
> ### W2.3
> For Table 2, the best entry is usually bolded. As for Figure 4b, I apologize for the imprecise comment—I can’t recall what I meant and don’t see any issue with it now.
>
> ### W3
> Thank you for that clarification. My comment was more about whether there are missing constraints on $d_f$ or $d_r$. Don’t you need to ensure that $d_f > n_f$ and that $d_r > n - n_f$ as well?

---

> > ### Author Response · Authors · 2024-11-26
> >
> > We sincerely thank you for your thoughtful comments and constructive feedback. We hope to address your concerns as follows:
> >
> > **Response to W1 / W2.2:**
> > In Table 2, the $\alpha$ values are selected based on the best performance for each method. For example, for CE, we use $\alpha=0.8$; for ICE, $\alpha=0.1$; and for $\mathrm{KL}, \alpha=0.2$. In Table 4, we provide each result with varying $\alpha$.
> >
> > **Response to W2.3:**
> > Thank you for pointing this out! We will ensure the best entries in Table 2 are bolded.
> >
> > **Response to W3:**
> > Thank you for the observation. We will include this assumption to clarify the constraints.
> >
> > Thank you once again for your valuable time and positive feedback. We greatly appreciate your efforts in helping us improve our work!

---

### Author Response · Authors · 2024-11-20
**General Response**

We sincerely thank all reviewers and chairs for their valuable time and efforts in providing detailed feedback on our work. In particular, we appreciate the insightful reviews and constructive suggestions from Reviewers bHDY, h2Yv, NAAU, and J9SB, as well as the positive feedback from Reviewers bHDY and J9SB.

Moreover, we would like to address some common concerns raised by the reviewers here:

**1. Why choose overparameterized linear models and focus on class-wise unlearning experiments?**

Overparameterized linear models are widely adopted as a foundational framework for studying learning problems (e.g., transfer learning [1], continual learning [2], In-context learning [3,4]) and can be extended to more general settings such as neural tangent kernel (NTK) analysis. While the setting of overparameterized linear regression is indeed simplistic, it provides a valuable starting point for analyzing training dynamics, capturing the trajectory of learning rather than just upper or lower bounds.
In this work, we start with the overparameterized linear regression model and hope to extend the analysis to more general cases such as multi-layer neural network in the future.

As for the focus on regression versus classification, although we use linear regression to analyze the fine-tuning process, our
assumptions about the data structure are based on a classification problem, which is why our experiments focus on class-wise forgetting.

**2. The novelty of the discriminative regularizer.**

We agree that there is limited distinction between our proposed regularization approach and existing objective functions, primarily differing in the choice of the loss function and the parameter $\alpha$.

However, the main novelty of our paper lies in providing the first theoretical framework to understand the empirical phenomenon of why naive fine-tuning fails to forget. Based on the analysis in Sections 3 and 4, we aim to address this phenomenon by designing suitable objectives. Additionally, Theorem 4.1 provides the rationale for selecting the parameter $\alpha \in(0,1]$ for the unlearning loss, whereas in [1], the constraint $\alpha \in(0,1]$ is applied to the retaining loss. **While our regularizer may appear similar to existing formulations, our novelty/contribution extends beyond the regularizer to the broader theoretical and empirical insights we provide.**

In response to the insightful comments from Reviewers h2Yv and NAAU, we have rewritten Section 5 to emphasize the redesign rationale for the regularizer and its implications, rather than introducing it. Section 6 further validates our design through experimental results.

Thank you again for your valuable feedback!

References:

[1] Wu, Jingfeng, et al. "The power and limitation of pretraining-finetuning for linear regression under covariate shift." Advances in Neural Information Processing Systems 35 (2022): 33041-33053.

[2] Ding, Meng, et al. "Understanding Forgetting in Continual Learning with Linear Regression." arXiv preprint arXiv:2405.17583 (2024).

[3] Wu, Jingfeng, et al. "How Many Pretraining Tasks Are Needed for In-Context Learning of Linear Regression?." arXiv preprint arXiv:2310.08391 (2023).

[4] Chen, Xingwu, Lei Zhao, and Difan Zou. "How transformers utilize multi-head attention in in-context learning? a case study on sparse linear regression." arXiv preprint arXiv:2408.04532 (2024).

---

### Meta-Review · Area_Chair_WLCs · 2024-12-23

**Metareview:**

Summary

This paper studies specially the fine-tuning problem in machine unlearning. As it is observed in many papers that fine-tuning using the retaining data only will not incur a good enough forgetting performance as the retrain, the paper explores further this phenomenon by using a simple linear regression model, and simulated simplified data, from both theoretical and empirical perspectives. The identified insights from these theoretical and empirical studies are that – when both retaining data and forgetting data are considered, more weight should be given to the retaining data. The paper then demonstrates empirically that such insights are effective.

Strengths

1. Reviewer(s) pointed out that the fine-tuning is efficient and does not require forgetting data, so it is worth exploring direction.
2. The paper provides some theoretical studies and insights.
3. Empirical results demonstrate the effective of the insights.

Weaknesses

1. The whole analysis and empirical study to draw the most important insights and conclusions regarding fine-tuning is based on an oversimplified model (linear regression) and an oversimplified dataset (on which it is easy to tell different classes can be related to different features). It is not convincing that such a result or insight drawn is practical enough.
2. The most important insight drawn is that more weight should be given to retaining loss instead of forgetting loss in machine unlearning. Such a conclusion is anti-intuition and is in contradiction with the main motivation of the paper that fine-tuning on retaining data sole is not effective in forgetting. This insight is claimed to be one of the most important contributions of the paper, but not sufficiently due to such contradiction exists – such contradiction may be an important observation (but the paper still needs to show that both losses are necessary but just one should be given fewer weights), or a result from the oversimplified study but not extendable to more practical cases. The analysis and study around this insight is not sufficient in the current paper.
3. The studying point of the paper is some empirical evidence – when the retaining error is zero, the forgetting is far below the gold model (retraining model). I believe the problem with fine-tuning still exists, but an empirical study forcing the retaining error to be zero could be too strict and potentially could lead to overfitting. A more practical demonstration of the fine-tuning issue is recommended.
4. While the conclusion of the paper still recommends using forgetting data, then the pure fine-tuning methods may not be worth studying from the beginning. The paper needs to find a better way to argue the meaning of studying the fine-tuning problem while forgetting data is finally used. If forgetting data is not used, then the fine-tuning problem is worth studying; but if the forgetting data is used, the story could be different – which has not been sufficiently discussed in the paper either.

Recommendation

I recommend rejecting the paper at the current phase, due to the unsolved issues concerning the oversimplified model and problem setting, and the insufficient arguments about the contradiction of the insight provided from the theoretical results (especially given that the insight comes from an oversimplified study). On the other hand, I believe if the arguments and insights are considered in a more comprehensive way, the direction is worth a study even with the oversimplified theoretical studies. The theoretical studies themselves are good to start from something simple.

**Additional Comments On Reviewer Discussion:**

Two reviewers give six to the paper while one of them is still concerned that the oversimplified theoretical study may not be sufficient. The other two reviewers gave three to the paper. One of them gave two rounds of very long discussion expressing the two rounds of remaining concerns, which are mainly in Weaknesses 1 and 2. This reviewer is also concerned about the novelty of the paper in methodology; I agree with the authors' argument that their main contribution was not on methodology and the provided insights (if sufficiently argued) provided the methodology novelty compared to existing methods. Another reviewer who gave three also expressed remaining concerns on the paper,  mainly in Weaknesses 3 and 4. This reviewer is also concerned that the oversimplified study cannot be extended.

While I agree with the authors that the simplified study is a good starting point for theoretical analysis, which is potentially challenging, I am concerned about the anti-intuition result provided, and the contradiction with the motivation of the paper that pure fine-tuning is not good. As said, – such contradiction may be an important observation (but the paper still needs to show that both losses are necessary but just one should be given less weight), or a result from the oversimplified study but not extendable to more practical cases. So I think the paper is not ready in the current phase, while agreeing that the direction is worth exploring from a theoretical point of view.

---

### Decision · Program_Chairs · 2025-01-22

Reject